

# A simplified model of precipitation enhancement over a heterogeneous surface

Guido Cioni[1,2,3] and Cathy Hohenegger[1,3]

[1]Max Planck Insitute for Meteorology, Hamburg, Germany
[2]International Max-Planck Research School on Earth System Modelling
[3]Hans Ertel Centrum for Weather Research

*Correspondence to:* Guido Cioni (guido.cioni@mpimet.mpg.de)

**Abstract.** Soil moisture heterogeneities through the triggering of mesoscale circulations influence the onset of convection and subsequent evolution of thunderstorms producing heavy precipitation. However local evaporation also plays a role in determining precipitation amounts. Here we aim at disentangling the effect of advection and evaporation on precipitation over the course of a diurnal cycle by formulating a simple conceptual model. The derivation of the model is inspired from the results

of simulations performed with a high-resolution (250 m) Large-Eddy Simulation model over a surface with varying degrees of heterogeneity. Key element of the model is the representation of precipitation as weighted sum of advection and evaporation, each weighted by its own efficiency. The model is then used to isolate the main parameters that control the variations of precipitation over spatially drier patches. It is found that these changes surprisingly do not depend on soil moisture itself but instead purely on parameters that describe the atmospheric initial state. The likelihood for enhanced precipitation over drier

soils is discussed based on these parameters. Additional experiments are used to test the validity of the model.

## 1 Introduction

Will more soil moisture lead to more or less precipitation? This apparently simple question inspired many studies over the course of the last 50 years. Over a homogeneous surface precipitation is expected to increase with surface evaporation, and thus with soil moisture in a soil moisture-limited regime (Manabe, 1969; Budyko, 1974), regardless of the atmospheric state

(Cioni and Hohenegger, 2017) as long as convection can be triggered on both dry or wet surfaces. However, the real world is far from being homogeneous. The presence of heterogeneity in soil moisture induces thermally-driven meso-scale circulations (Segal and Arritt, 1992) which transport moist air from spatially wetter patches to spatially drier patches, acting against the initial perturbation of soil moisture, and which can affect the distribution of precipitation.

Many idealized studies investigated the effect of such circulation on convection and ensuing precipitation. Avissar and Liu

(1996) found that land-surface wetness heterogeneity (i.e. spatial gradients of soil moisture) controls the transition from a randomly scattered state of convection to a more organized one where clouds form at the front of the meso-scale circulation. The presence of such circulations also tend to enhance the precipitation amount. Further analyses have shown that this basic response can be modified by many environmental factors.

Yan and Anthes (1988) found that accumulated precipitation over spatially dry patches is maximized when the patch length is





comparable to the local Rossby radius of deformation ($\sim$ 100 km in mid-latitudes), a result that was later confirmed by Chen and Avissar (1994) and Lynn et al. (1998). Robinson et al. (2008) proposed an alternative explanation by which the effect of surface hot spots is maximized for wavelength of roughly 50 km when the aspect ratio of the applied heating matches the ratio of vertical and horizontal wavenumbers demanded by the dispersion relation for buoyancy (gravity) waves. Froidevaux

et al. (2014) explored the interaction between horizontal soil moisture variations, wind and precipitation. They found that, only when winds are too weak to control the propagation of thunderstorms more precipitation is observed over drier surfaces. Finally the response of precipitation also depends upon the background atmospheric profile. Chen and Avissar (1994) found that the presence of a moist atmospheric profile over a spatially drier surface reduces the precipitation advantage as the surface fluxes of moisture are reduced. Hence, from such studies, an increase of precipitation over spatially drier patches is expected when

the gradient of surface wetness is high, the soil moisture heterogeneity length-scale is around 50-100 km and no background wind is present.

These same mechanisms can be observed in some areas of the world, the so-called hot spots of land-atmosphere interactions (Koster et al., 2004). Several observational studies (e.g. Taylor et al. (2012)) showed that in the Sahel region thunderstorms occur preferably over regions drier than their surroundings. In other areas of the world the synoptic forcing is usually so strong

that a robust relationship of causality between soil moisture and precipitation can not be found (Tuttle and Salvucci, 2017). Instead of speaking of heterogeneous or homogeneous conditions, Guillod et al. (2015) have indicated that over most areas of the world, except the Sahel, a negative spatial coupling coexists together with a positive temporal coupling.

Although the aforementioned studies have qualitatively shown how precipitation is influenced by soil moisture, soil moisture gradients and by the atmospheric environment, here we aim at developing a simplified conceptual model to isolate under which

conditions an increase in precipitation is more likely. We focus on the case of a heterogeneous surface. In this case precipitation is not only affected by the advection of moisture due to the meso-scale circulation but also by local evaporation (Wei et al., 2016). These two factors depend differently on soil moisture.

The meso-scale circulation triggered by the surface wetness heterogeneity strengthens with decreasing local soil moisture, as this gives a larger spatial gradient of surface fluxes and thus of surface pressure. Instead local evaporation is limited with

reduced local soil moisture. The superposition of local evaporation and remote moisture advection eventually contribute to the observed precipitation, with the atmosphere being the medium that weights these two different contributions.

We thus need to quantify two different effects. First, how local evaporation as well as moisture advection depend on soil moisture. Second, what is the relative role of the atmosphere, or in other words the efficiency in converting these potential moisture sources into precipitation. The results obtained by the aforementioned studies seem to suggest that a local increase

of local soil moisture will lead to a negative variation of precipitation, i.e. the derivative of precipitation with respect to soil moisture is negative. Our goal is to find a simple formulation for the derivative of precipitation as function of soil moisture in case of an heterogeneous surface.

Lintner et al. (2013) already derived an equation for the derivative of precipitation with respect to soil moisture based on a model of intermediate-level complexity of the tropical atmosphere (Quasi-equilibrium Tropical Circulation Model 1 (QTCM1),

Neelin and Zeng (2000)). Inspired by their work, we develop a theoretical model which is based on similar assumptions but





greatly simplifies the formulation of moisture advection and evaporation which are all expressed in terms of linearized features. Also the fact that we consider the specific case of advection by a meso-scale circulation and not by the large-scale flow, as in Lintner et al. (2013), will allow us to simplify the idealized framework.

We aim here at a minimal representation that allows us to isolate what are the fundamental quantities that cause variations
of precipitation with soil moisture and to determine which is the most efficient way to increase precipitation with local soil moisture. To this aim in section 2.1 we build an idealized framework that allow us to simulate the evolution of convective clouds and precipitation over land during a diurnal period. After a brief analysis of the features of the convective diurnal cycle in section 3.1 we estimate the various terms of the moisture balance and in particular the efficiencies of the conversion of evaporation and advection into precipitation in section 3.2. By deriving a simple conceptual model that agrees with the results
of model simulations in section 4 we will show that, at least to a first order, the change of precipitation with soil moisture does not depend on the soil moisture content itself and that the most efficient way to increase precipitation consists in increasing the surface wetness gradient. The results are concluded in section 5.

## 2 Method

### 2.1 Experimental design

The modelling framework used in this work is, in terms of physical parametrizations and dynamical core, identical to the one described in Cioni and Hohenegger (2017), to which the reader is referred for details. We use the ICOsahedral Non-hydrostatic - Large Eddy Model (ICON-LEM) as atmospheric model coupled to the land-surface model, TERRA-ML, to simulate the diurnal cycle of convection over idealized land surfaces from 6 Local Standard Time (LST) to 24 LST.

The horizontal periodic domain spans $1600 \times 400$ points with a grid spacing of 250m, which results in a size of approximately
(given the triangular grid) $400 \times 100$ km$^2$. In the vertical dimension 150 levels are distributed from the surface up to the model top located at 21 km: the spacing reaches 20 m in the lower levels and 400 m close to the model top. In contrast to Cioni and Hohenegger (2017), heterogeneous surface conditions are considered. The heterogeneity is prescribed by dividing the x-direction of the domain into two patches having the same surface area of $200 \times 100$ km$^2$. Figure 1 displays a sketch of the domain setup, together with a visual representation of convective features that will be discussed later.

The domain is rectangular in order to limit computational expenses and is elongated in the $x$-axis given that the front associated with the simulated meso-scale circulation is expected to sweep this axis. Note that the chosen patch size of 200 km is larger than the optimal values identified by Chen and Avissar (1994); Yan and Anthes (1988); Lynn et al. (1998), which, in this sense, reduces the dynamic contribution of advection on precipitation. The sensitivity to different $y$-axis size was also tested, and resulted in a negligible influences. The patch size was varied in sensitivity experiments and is discussed when needed along
the text.

The introduced heterogeneity is set by using two different values for the volumetric soil moisture $\phi$ [m$^3$ m$^{-3}$] over the two patches. At the initialization time the soil moisture of both the dry patch, $\phi_{dry}$, and of the wet patch, $\phi_{wet}$, are set constant overt the entire vertical soil depth to ease the interpretation of results. The other parameters that characterize the land surface,



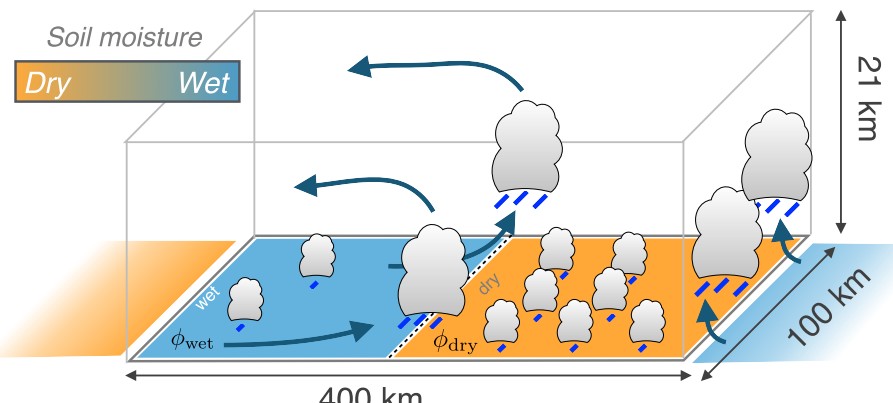

**Figure 1.** Sketch of the employed experimental framework. The initial condition for soil moisture and the expected initial development of convection are also sketched in order to ease the interpretation of the results.

including the soil temperature, are the same for the entire domain. Initially the soil temperature is prescribed using a linear profile which includes a climatological layer with a temperature of 281 K and a surface layer which has the same temperature as the overlying lowermost level of the atmosphere (see Cioni and Hohenegger (2017)).

Accordingly, the atmospheric initial state is spatially homogeneous except for random perturbations added to the vertical veloc-

5  ity and the virtual potential temperature in the lowermost 3 levels to break the perfectly symmetric initial state. The atmosphere is initialized using the dry-soil advantage profile of Findell and Eltahir (2003), which corresponds to a sounding observed on 23 July 1999 in Lincoln, Illinois, USA. However, we set winds to zero all over the atmospheric column to simplify the analysis. We perform a set of experiments by setting at the initial time $\phi_{\mathrm{wet}}$ to the saturation value and varying $\phi_{\mathrm{dry}}$, with values ranging from the saturation to 20% of the saturation value, which is below the wilting point for the chosen soil type. The upper part of

10  Tab. 1 summarizes the simulations performed with this basic configuration.

In order to test the validity of the theory proposed in section 2, based on this set of basic experiments, we perform further sensitivity experiments. First, we decrease the initial value of $\phi_{\mathrm{wet}}$ to 70 % of the saturation value. Second, we change the initial atmospheric profile to the one used in Schlemmer et al. (2012) which represents an idealization of the typical atmospheric state prone to convection in Europe. Results of these perturbed experiments will be briefly described throughout the paper when

15  needed.





| Name | Sounding | $\phi_{dry}$ | $\phi_{wet}$ |
|---|---|---|---|
| Basic configuration | | | |
| DA_20_100 | DA | 20 | 100 |
| DA_30_100 | | 30 | 100 |
| DA_40_100 | | 40 | 100 |
| DA_50_100 | | 50 | 100 |
| DA_60_100 | | 60 | 100 |
| DA_65_100 | | 65 | 100 |
| DA_70_100 | | 70 | 100 |
| DA_80_100 | | 80 | 100 |
| DA_100_100 | | 100 | 100 |
| Not-saturated wet patch | | | |
| DA_20_70 | DA | 20 | 70 |
| DA_30_70 | | 30 | 70 |
| DA_40_70 | | 40 | 70 |
| DA_50_70 | | 50 | 70 |
| DA_60_70 | | 60 | 70 |
| DA_70_70 | | 70 | 70 |
| Idealized sounding | | | |
| ID_20_100 | ID | 20 | 100 |
| As in basic | | | |
| ID_100_100 | ID | 100 | 100 |

**Table 1.** Overview of the performed simulations. The first column indicates the name, while the second column indicates the sounding used for initialization: DA for dry soil advantage after Findell and Eltahir (2003) and ID for idealized after Schlemmer et al. (2012). Third and fourth columns indicate the value of soil moisture over the dry $\phi_{dry}$ and wet $\phi_{wet}$ patches in percentage of the saturation values. Naming convention for the experiments follows SOUNDING_$\phi_{dry}$_$\phi_{wet}$.

## 3 Results

### 3.1 General features of convection

The differential heating of the two patches, caused by the heterogeneity in soil moisture, manifests itself in a gradient of both sensible and latent heat fluxes. Here, we describe the general features of the extreme case, DA_20_100, which, in a nutshell,

5 reproduces the features expected from this kind of simulations. At 12 LST the difference in sensible heat fluxes between the two patches reaches almost 280 W m$^{-2}$. This results in a difference in near-surface virtual potential temperature of about 4 K at the same time. As a consequence, a pressure gradient of about 1 hPa develops close to the surface, which supports a thermally-driven circulation (Segal and Arritt, 1992). The circulation is constituted by a front of moist air moving inland over





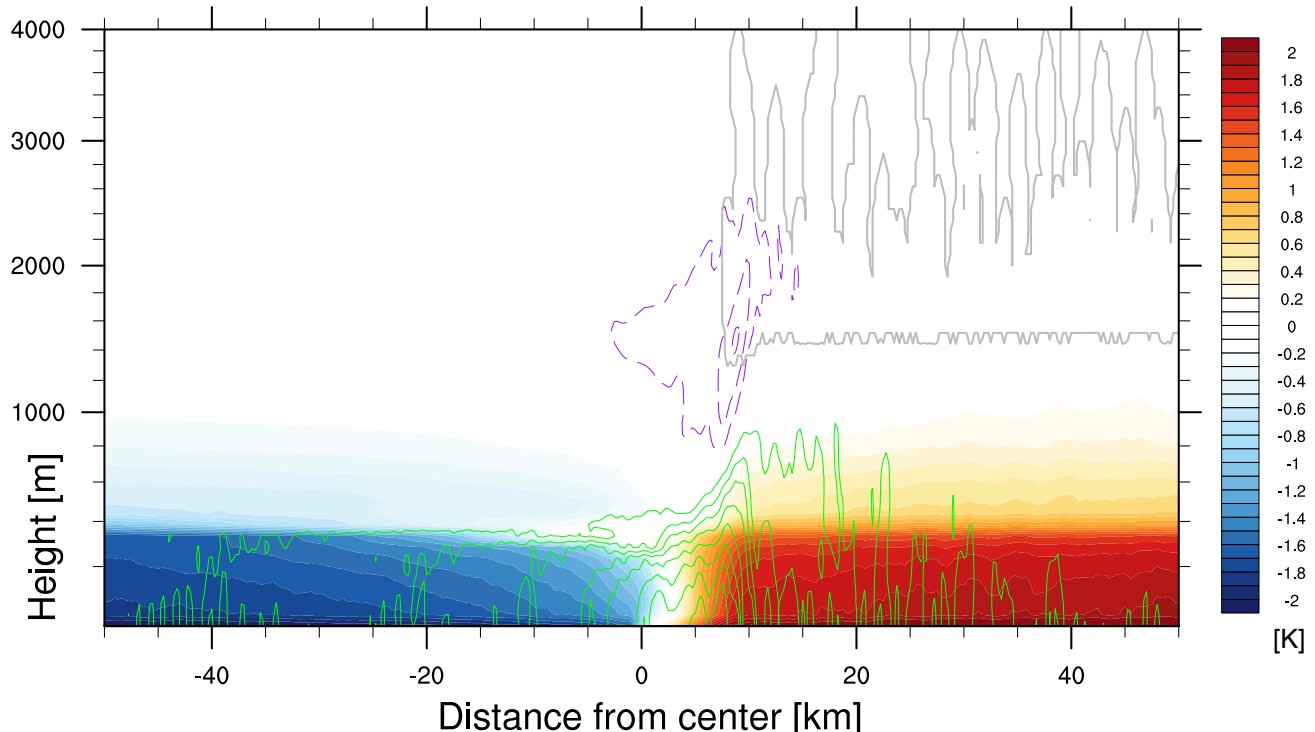

**Figure 2.** $x$-$z$ diagram at 1200 LST of $y$-averaged quantities for the `DA_20_100` case. Temperature anomaly (color contours), zonal winds (contour lines from -4 to 4 every 0.5 m s$^{-1}$, positive values with green contours, negative values with purple ones) and cloud water mixing ratio (grey contour, only $10^{-5}$ g kg$^{-1}$ isoline). On the $x$-axis numbers indicate the distance from the center of the domain in km. On the $y$-axis height from the surface is indicated in meters.

the dry patch at lower levels (from the surface up to 1 km) and a return flow between 1 and 3 km, as shown by the wind contours in Fig. 2. As a result of the circulation, and as found in past studies, convection preferentially develops over the dry patch and in particular on the border between the wet and the dry patch.

In order to track the front associated with the circulation we use an algorithm designed to follow the front only over the dry

5 patch. The algorithm is based on the $y$-averaged zonal wind speed at approximately 100 m of height and is triggered when the velocity in the middle of the domain reaches 1 m s$^{-1}$ and automatically interrupted when the opposite fronts collide. At every time instant a search of the maximum value of zonal wind speed is performed in a box centered on a first guess based on a linear extrapolation of the previous time instants. This is necessary in order to maintain the focus of the tracking algorithm on the edge of the front. The size of the box is the only parameter that needs to be tuned when tracking the front in different

10 simulations. Otherwise, the algorithm is robust.

Figure 3 (a) shows the Hovmöller diagram of the zonal wind and the tracked position of the front every 15 minutes with shaded circles for the case `DA_20_100`. In Fig. 3 (b) the position and speed of the front obtained with the aforementioned algorithm



| Case | $\phi_{\mathbf{dry}}$ | $A_{\mathbf{dry}}$ | $E_{\mathbf{dry}}$ | $P_{\mathbf{dry}}$ | $\eta$ |
|------|------|------|------|------|------|
| DA_20_100 | 0.0908 | 7.796 | 0.0008 | 1.255 | 0.161 |
| DA_30_100 | 0.1362 | 7.467 | 0.0113 | 1.077 | 0.144 |
| DA_40_100 | 0.1816 | 7.223 | 0.1140 | 1.005 | 0.137 |
| DA_50_100 | 0.2270 | 6.673 | 0.6373 | 0.912 | 0.125 |
| DA_60_100 | 0.2724 | 4.665 | 2.0271 | 0.888 | 0.133 |
| DA_65_100 | 0.2951 | 3.222 | 3.0393 | 0.805 | 0.129 |
| DA_70_100 | 0.3178 | 1.734 | 4.0920 | 0.708 | 0.122 |
| DA_80_100 | 0.3632 | -0.412 | 5.2770 | 0.533 | 0.094 |
| DA_100_100 | 0.4540 | 0.031 | 5.0800 | 0.560 | 0.110 |

**Table 2.** Values of soil moisture, advection, evaporation and precipitation over the dry patch accumulated over the diurnal cycle. All values have mm units. The rightmost column shows the precipitation efficiency computed as $\frac{P_{\mathrm{dry}}}{A_{\mathrm{dry}}+E_{\mathrm{dry}}}$.

are displayed. The front starts to slowly propagate in the late morning with a velocity smaller than $2\,\mathrm{m\,s^{-1}}$ but is later accelerated by cold pools, in agreement with Rieck et al. (2015). The cold pools are formed after the first strong precipitation event. The speed of the front reaches values larger than $8\,\mathrm{m\,s^{-1}}$ before the front collides with the opposing front coming from the outer boundary due to the periodic domain. When the soil moisture of the dry patch exceeds 70% of the saturation value no circula-

tion forms, because the gradient in surface temperature is too weak to cause a pressure difference between the patches. In this case the convection transitions to a randomly scattered state (Avissar and Liu, 1996) and we define the speed of the front to be 0.

## 3.2 Local and remote sources of precipitation

The diurnal cycle of precipitation can be inspected and compared to the one of evaporation and advection, using the method-
ology introduced in appendix A. Figure 4 shows the various components of the moisture balance computed from the model output every 5 minutes averaged over the dry patch as well as over the entire domain. It is immediate to verify that the advection term averaged over the entire domain is zero, as expected. Instead, when considering the residual averaged over the dry patch this term is always positive, indicating a net transport of moisture from the wet to the dry patch.

The advection of moisture over the dry patch increases in the late morning as a result of the increasing thermal gradient be-
tween the two patches and reaches a maximum around 13 LST. This behaviour is similar to the one observed by Yan and Anthes (1988, Fig. 9). The first deep convection event in DA_20_100 produces a strong cold pool which causes a strong surface divergence, i.e. the minimum at about 14 LST in Fig. 4. Given that the maximum of precipitation associated with this event is located in the vicinity of the boundary between the wet and the dry patch, this induces a net negative effect on $A_{\mathrm{dry}}$.

In order to study the variation of the moisture budget terms as function of $\phi_{\mathrm{dry}}$ we conduct the same moisture balance analysis
for every simulation and integrate the values over the entire diurnal cycle (18 hours). Results are reported in Tab. 2. As expected





the advection term decreases with increasing local soil moisture whereas local evaporation increases. Overall the accumulated precipitation averaged over the dry patch decreases when the soil moisture increases, as shown also in Fig. 5. This already suggests that advection appears to be more efficient than evaporation in producing precipitation, as the increase of $E_{\mathrm{dry}}$ with soil moisture is not followed by an increase in $P_{\mathrm{dry}}$.

These qualitative observations can be formalized by defining precipitation efficiencies. This approach was firstly proposed by Budyko (1974) and later adopted by many studies including the one of Schär et al. (1999). The overall assumption underlying the pioneering work of Budyko (1974) is that moisture coming from inside (local evaporation) or outside (remote advection) of some closed domain is well mixed. Under this assumption one can express the precipitation over a certain area as:

$$P_{\mathrm{area}} = \eta(A_{\mathrm{area}} + E_{\mathrm{area}}) \tag{1}$$

where $\eta$ is the precipitation efficiency. All the terms are considered as areal averages and integrated over a certain time period. The rightmost column of Tab. 2 shows the efficiency $\eta$ computed according to Eq. 1. It can be seen that, in this case, convection is not so efficient in converting local and remote sources of moisture into precipitation as the values range from 16% to 11%. This range of values is interpreted as resulting from the different processes that contribute to precipitation over the dry patch. In fact, in the case `DA_20_100`, evaporation over the dry patch is negligible, i.e. $E_{\mathrm{dry}} \simeq 0$, so that Eq. 1 applied to the dry patch

reads $P_{\mathrm{dry}} \simeq \eta A_{\mathrm{dry}}$. Thus, the efficiency obtained in this case is representative of the advection process and can be interpreted as an *advection efficiency* $\eta_{\mathrm{A}}$. On the other hand, in `DA_100_100` the advection is negligible so that in this case we obtain an *evaporation efficiency* $\eta_{\mathrm{E}}$. Taking all these findings together, we rewrite Eq. 1 as:

$$P_{\mathrm{area}} = \eta_{\mathrm{A}} \cdot A_{\mathrm{area}} + \eta_{\mathrm{E}} \cdot E_{\mathrm{area}} \tag{2}$$

where now $\eta_{\mathrm{A}} \neq \eta_{\mathrm{E}}$.

Fig. 5 confirms that, regardless of the particular choice of the efficiency $\eta$, the decrease of precipitation over wetter soils cannot be captured when using a single efficiency. Instead, using the two efficiencies gives a much better match with the simulated value of $P_{\mathrm{dry}}$ (see yellow line in Fig. 5). Also by using two efficiencies, the latter become independent of soil moisture. The resulting efficiencies, obtained through a fit of Eq. 2, are $\eta_A = 0.15$ and $\eta_E = 0.10$.

The fact that one efficiency doesn't match well, in contrast to previous studies, may be linked to the fact that we consider a

small domain and a short time scale. The assumption of a well-mixed atmosphere is likely to hold better on a continental (e.g. Europe) and seasonal scale, as in Schär et al. (1999). Using two efficiencies nevertheless as it requires data from at least two simulations to estimate them.

Initializing the atmosphere with a different sounding will likely lead to different efficiencies. This is illustrated with the `ID_` cases (see Tab. 3), where another sounding was used to initialize the atmosphere (see section 2). The efficiencies computed

for this case range from 47% to 35%, indicating that the atmosphere is more efficient at converting advection and evaporation into precipitation. Nonetheless, also for this case $\eta_A > \eta_E$. Evaporation now reaches a maximum which is larger than the one attained by advection. This is mainly an effect of larger precipitation amounts that fall over the wet patch which in turn prevents an efficient advection of moisture from the wet to the dry patch.



| Case | $A_{\mathbf{dry}}$ | $E_{\mathbf{dry}}$ | $P_{\mathbf{dry}}$ | $\eta$ |
|---|---|---|---|---|
| ID_20_100 | 3.814 | 0.008 | 1.789 | 0.468 |
| ID_30_100 | 3.861 | 0.028 | 1.912 | 0.492 |
| ID_40_100 | 3.920 | 0.143 | 1.671 | 0.411 |
| ID_50_100 | 3.557 | 0.659 | 1.740 | 0.413 |
| ID_60_100 | 2.631 | 2.054 | 1.759 | 0.376 |
| ID_70_100 | 0.865 | 4.080 | 1.542 | 0.312 |
| ID_80_100 | -0.068 | 4.884 | 1.652 | 0.334 |
| ID_100_100 | 0.022 | 4.776 | 1.662 | 0.346 |

**Table 3.** As in Tab. 2 but for the additional sounding.

In the ID_ cases the decrease of precipitation observed over the dry patch with higher values of soil moisture is smaller than the one obtained in the DA_ cases. Although this could be related to a weaker sensitivity of that particular atmospheric state to the meso-scale circulation, we note that the amount of precipitation computed strongly depends on the collision of the fronts. As shown in Fig. 6 the collision of the fronts in the center of the dry patch has different effects on precipitation depending on the atmospheric state. In the ID_20_100 case strong precipitation events with local maxima of 10 mm h$^{-1}$ are produced in the center of the patch after the front collision and several secondary events develop due to the waves propagating away from the collision. Instead, in the DA_20_100 case no strong precipitation event is produced when the fronts collide.

## 4 Theoretical model

In section 3.2 we showed that precipitation can be expressed as a linear combination of advection and evaporation weighted by different efficiencies which are assumed independent of soil moisture. Knowing this we can now try to answer one of the first question that was posed in the introduction: What are the minimum parameters that control the variation of precipitation with soil moisture? In order to do so we first have to derive some functional forms of evaporation and advection in terms of soil moisture.

### 4.1 Surface evaporation

The simplest parametrization of evaporation (we will neglect the transpiration part given that our study do not consider plants) is the so-called *bucket model* introduced by Budyko (1961) and extended by Manabe (1969). Evaporation is defined as a potential term controlled by a limiting factor (also called stress factor). Here we use such formulation to approximate the surface latent





heat flux $\mathrm{LH}(\phi, t)$ as

$$\mathrm{LH}(\phi, t) = \mathcal{A} Q_{\mathrm{net}}(t) \times \begin{cases} 0 & \text{for } \phi < \phi_{\mathrm{wp}} \\ \frac{\phi - \phi_{\mathrm{wp}}}{\phi_{\mathrm{crit}} - \phi_{\mathrm{wp}}} & \text{for } \phi_{\mathrm{wp}} < \phi < \phi_{\mathrm{crit}} \\ 1 & \text{for } \phi > \phi_{\mathrm{crit}} \end{cases} \tag{3}$$

where $Q_{\mathrm{net}}(t)$ is the net incoming radiation at the surface (long-wave+short-wave), $\phi_{\mathrm{wp}}$ the soil permanent wilting point and $\phi_{\mathrm{crit}}$ is the critical threshold at which evaporation does not increase any more with increasing soil moisture. As explained by

Seneviratne et al. (2010) this does not usually correspond to the field capacity. The enthalpy of vaporization does not show up in Eq. 3 as units are already in mm h$^{-1}$.

$\mathcal{A}$ is a proportionality constant which needs to be introduced given that, even in the extreme case of a saturated soil, non-zero sensible heat fluxes and ground heat flux prevents the entire conversion of $Q_{\mathrm{net}}$ into latent heat fluxes. The constant $\mathcal{A}$ clearly depends on the particular soil model employed as well as on the different parameters that characterize the particular soil type

considered (e.g. albedo, heat capacity) and partially also on the atmosphere.

In order to link Eq. 3 to the accumulated evaporation $E_{\mathrm{dry}}$ contained in Eq 2 we can consider the time average of $Q_{\mathrm{net}}$, which we indicate as $\langle Q_{\mathrm{net}} \rangle$, and multiply for a time scale which is defined to be $\tau=18$ hours, that is the entire length of the simulation (and of the resulting diurnal cycle). The resulting Eq. 4 has mm units.

$$E_{\mathrm{dry}}(\phi) = \tau \mathcal{A} \langle Q_{\mathrm{net}} \rangle \times \begin{cases} 0 & \text{for } \phi < \phi_{\mathrm{wp}} \\ \frac{\phi - \phi_{\mathrm{wp}}}{\phi_{\mathrm{crit}} - \phi_{\mathrm{wp}}} & \text{for } \phi_{\mathrm{wp}} < \phi < \phi_{\mathrm{crit}} \\ 1 & \text{for } \phi > \phi_{\mathrm{crit}} \end{cases} \tag{4}$$

Note that in Eq. 4 instead of using $\langle \phi \rangle$ we just consider the value of soil moisture at the initialization time for the sake of simplicity. Equation 4 can now be used to fit the values of $E_{\mathrm{dry}}$ computed from simulations (Tab. 2) and obtain an unambiguous value for the parameters $\mathcal{A}$ and $\phi_{\mathrm{wp,crit}}$ (see Fig. 7). These are estimated to be $\mathcal{A} \simeq 0.663$, $\phi_{\mathrm{wp}} \simeq 0.213$ m$^3$m$^{-3}$ and $\phi_{\mathrm{crit}} \simeq 0.350$ m$^3$m$^{-3}$. Note that the latter estimate is not far from the field capacity of this soil type, i.e. 0.340 m$^3$m$^{-3}$, while the estimated wilting point is almost double the expected one. This is related to the fact that the employed bare soil evaporation scheme

tends to shut down evaporation too early as noted by Schulz et al. (2016) and Cioni and Hohenegger (2017). Thus, both $\phi_{\mathrm{wp,crit}}$ depend not only on the particular soil type employed but also on the soil model.

Figure 7 shows the fit of Eq. 4, together with the values obtained in the simulations. It reveals a very good agreement between theory and simulations. The small discrepancies mainly come from the fact that we assume a constant value of $\langle Q_{\mathrm{net}} \rangle = 300$ W m$^{-2} = 0.43$ mm h$^{-1}$, although the simulated value depends on soil moisture and varies by about 7%. This is due to different

cloud regimes which alter the surface radiation balance (Cioni and Hohenegger, 2017, Sec. 4b).

We note that our formulation of evaporation differs from the one used in the model of Lintner et al. (2013) where potential evaporation was used in place of $Q_{\mathrm{net}}$. Our approach in deriving the theoretical model directly neglects the feedback between the land-surface and the atmosphere, as evaporation does not implicitly depend on the near-surface atmospheric specific humidity but only on soil moisture.





## 4.2 Advection

$A_{\text{dry}}$ can be obtained directly from its mathematical definition assuming that the advection of every tracer is mainly due to the propagation of the breeze front (see Appendix A for an exhaustive definitions of the symbols) :

$$A_{\text{dry}} = \frac{1}{\rho_w} \tau \int_{H_{\text{front}}} \rho_a \left\langle \nabla (q_{\text{tot}} \cdot u_{\text{front}}) \right\rangle \mathrm{d}z \tag{5}$$

where $H_{\text{front}}$ is the height of the front associated with the meso-scale circulation or, equally, the Planetary Boundary Layer (PBL) height, $u_{\text{front}}$ its speed, $\rho_a$ is the air density and $\rho_w$ the water density. A fundamental approximation is already applied when writing Eq. 5 and should be highlighted. The features that describe the front propagation, like its speed, were defined only up to the collision time (see e.g. Fig. 3). Thus in Eq. 5 we are implicitly neglecting the contributions that may arise after the front collision. We are aware that one may solve this issue by considering the variables $E_{\text{dry}}$, $A_{\text{dry}}$ and $P_{\text{dry}}$ accumulated

only up to the front collision. However this would possibly introduce different errors in the evaluation of the aforementioned variables as the time of collision would need to be defined and is not always clearly visible (see again Fig. 6).

Eq. 5 is further approximated by:

$$A_{\text{dry}} \sim \tau \frac{\rho_a}{\rho_w} \frac{H_{\text{front}}}{L_{\text{front}}} \left\langle \Delta \overline{q}_{\text{tot}} \cdot u_{\text{front}} \right\rangle \tag{6}$$

With $\Delta \overline{q}_{\text{tot}}$ we indicate the difference in the vertical integral of $q_{\text{tot}}$ between the two patches. Note that, although we use the

same notation as in Appendix A, this time $\overline{q}_{\text{tot}}$ is integrated only up to $H_{\text{front}}$, which is considered constant over the time when the front is active. Furthermore, in deriving Eq. 6 we assumed that the horizontal gradient of $q_{\text{tot}}$ depends only on the $x$-dimension and is constant in the vertical dimension between the surface and $H_{\text{front}}$.

Because of its definition, $\Delta \overline{q}_{\text{tot}}$ entails the contributions of every tracer. To simplify the problem we assume $\Delta \overline{q}_{\text{tot}} \simeq \Delta \overline{q}_v$, which is viewed as the difference in humidity ahead of the front and behind it. As in studies which have viewed sea breezes as

gravity current (Robinson et al., 2013), we assume that this difference is not directly affected by the circulation, which gives an upper bound estimate. Hence we can write:

$$q_v(t) = q_v(0) + \frac{\rho_w \text{LH}}{\rho_a H_{\text{moist}}} \tau_{\text{moist}} \tag{7}$$

where $q_v(0)$ is the specific humidity at the initial time, $H_{\text{moist}}$ is the vertical extent of the moistening process due to the latent heat flux LH and $\tau_{\text{moist}}$ its time scale. In computing the difference in specific humidity between the two patches $\Delta q_v$ the term

$q_v(0)$ disappears, as the initial condition is homogeneous. Thus, by assuming that the moistening is confined to the PBL and that $\tau_{\text{moist}} = \tau$ we can rewrite the advection as

$$A_{\text{dry}} \sim \tau \frac{\tau \left\langle u_{\text{front}} \right\rangle}{L_{\text{front}}} \left\langle \Delta \text{LH} \right\rangle \tag{8}$$

Regarding the other terms of Eq. 8, the results of the simulations show that the front has a constant inland propagation of $L_{\text{front}} \simeq$ 100 km, which corresponds to half of the patch size. More importantly, the front speed does not vary much with different surface

heterogeneity gradients, against our initial expectations that motivated this study (see Introduction). For example, between the





`DA_20_100` and the `DA_60_100` cases only a 3% relative decrease in the front speed is observed (not shown).

This counterintuitive behaviour is related to the fact that cold pools cause the first noticeable acceleration of the front, as seen in section 3.1.This stands in agreement with what found by Rieck et al. (2015), and in particular with the thermodynamic contribution of cold pools to the propagation speed of the front (their Eq. 1). In our case cold pools are distributed along the

front and continuously fed by precipitation events behind of it similarly to what happens in squall-lines. Given this spatial organization, their strength and propagation does not depend on the surface state as in Gentine et al. (2016) but mostly on the upper air state as in Peters and Hohenegger (2017), the latter one not being greatly modified by surface fluxes during the diurnal cycle.

We can thus finally express advection simply as

$$A_{\mathrm{dry}}(\phi) = \tau \mathcal{B} \langle \Delta \mathrm{LH}(\phi) \rangle \tag{9}$$

where $\mathcal{B} = \tau \langle u_{\mathrm{front}} \rangle / L_{\mathrm{front}}$ is a proportionality constant that does not depend on the surface state. The difference $\langle \Delta \mathrm{LH} \rangle$ is known, thanks to the previous section, and can be used to fit the values of advection obtained in the simulations with Eq. 9 to obtain a value of the parameter $\mathcal{B} = 1.47$. This is smaller than the value that would be obtained by estimating instead $\langle u_{\mathrm{front}} \rangle$ and $L_{\mathrm{front}}$ directly, as this latter approximation does not take into account moisture losses due to advection. Note that from Eq.

9 it immediately follows that $A_{\mathrm{dry}}(\phi) = \mathcal{B} \Delta \mathrm{E}(\phi)$.

Figure 8 shows the values of $A_{\mathrm{dry}}$ and the fit performed using Eq. 9 for the basic set of experiments and for further cases, the latter used to test the findings that $\mathcal{B}$ does not depend on $\phi$. Overall the fit matches the variation of $A_{\mathrm{dry}}$ with $\phi_{\mathrm{dry}}$ remarkably well given the various assumptions. Both the simulated decrease of $A_{\mathrm{dry}}$ with higher values of soil moisture and the flattening of advection with soil moisture lower than the wilting point are reproduced, although both effects seem to be overestimated by

Eq. 9.

In the simulations where the initial value of $\phi_{\mathrm{wet}}$ is reduced to just 70% of saturation the estimated value of $\mathcal{B}$ is almost the same as the one of the default configuration, confirming that $\mathcal{B}$ does not depend both on $\phi_{\mathrm{wet}}$ and $\phi_{\mathrm{dry}}$. Instead, in the `ID_` cases (Tab. 3), which uses a different atmospheric profile and hence support distinct cold pool strength, the value of $\mathcal{B}$ is reduced by about half.

## 4.3 Computing the derivative of precipitation

Equations 2, 4 and 9 can be combined in order to compute $P_{\mathrm{dry}}$. We are, however, interested in its variation with soil moisture, $\frac{\partial P_{\mathrm{dry}}}{\partial \phi_{\mathrm{dry}}}$, which can be computed as:

$$\frac{\partial P_{\mathrm{dry}}}{\partial \phi_{\mathrm{dry}}} = \eta_A \frac{\partial A_{\mathrm{dry}}}{\partial \phi_{\mathrm{dry}}} + \eta_E \frac{\partial E_{\mathrm{dry}}}{\partial \phi_{\mathrm{dry}}}$$
$$= -\eta_A \mathcal{B} \frac{\partial E_{\mathrm{dry}}}{\partial \phi_{\mathrm{dry}}} + \eta_E \frac{\partial E_{\mathrm{dry}}}{\partial \phi_{\mathrm{dry}}} \tag{10}$$





Note that the derivation of Eq. 9 retains only one term of the difference given that $E_{\text{wet}}$ does not depend on $\phi_{\text{dry}}$.

Using Eq. 4 it is straightforward to compute the derivative of $E_{\text{dry}}$ as

$$\frac{\partial E_{\text{dry}}}{\partial \phi_{\text{dry}}} = \tau \mathcal{A} \langle Q_{\text{net}} \rangle \times \begin{cases} 0 & \text{for } \phi < \phi_{\text{wp}} \\ (\phi_{\text{crit}} - \phi_{\text{wp}})^{-1} & \text{for } \phi_{\text{wp}} < \phi < \phi_{\text{crit}} \\ 0 & \text{for } \phi > \phi_{\text{crit}} \end{cases} \tag{11}$$

which is a step-wise function constituted by constant values.

Equations 10 and 11 indicate that for $\phi < \phi_{\text{wp}}$ and $\phi > \phi_{\text{crit}}$ there is no change in precipitation with soil moisture independently of the value of the efficiencies. In contrast for $\phi_{\text{wp}} < \phi < \phi_{\text{crit}}$ then $\frac{\partial E_{\text{dry}}}{\partial \phi_{\text{dry}}} \neq 0$ and the equation can be rewritten to study the sign of $\frac{\partial P_{\text{dry}}}{\partial \phi_{\text{dry}}}$.

$$\frac{\partial P_{\text{dry}}}{\partial \phi_{\text{dry}}} \lessgtr 0 \Leftrightarrow -\eta_A \mathcal{B} + \eta_E \lessgtr 0 \Leftrightarrow \eta_E \lessgtr \eta_A \mathcal{B} \tag{12}$$

Putting the values of the efficiencies and of $\mathcal{B}$ obtained from the `DA_` simulations in the previous section in Eq. 12 confirms that, in that set-up, $\frac{\partial P_{\text{dry}}}{\partial \phi_{\text{dry}}} < 0$, which agrees with the simulations results. For the `ID_` cases, Eq. 12 leads to a similar result, although the difference $\eta_E - \eta_A \mathcal{B}$ is just slightly positive. We will explain why hereinafter.

Our simple theoretical derivation indicates that the variation of precipitation with soil moisture surprisingly does not depend on soil moisture itself. Instead, the atmospheric conditions, through the terms $\eta_A, \eta_E$ and $\mathcal{B}$, determine whether increasing or decreasing the soil moisture of the dry patch is needed to increase the precipitation amount. These findings contrast with the ones of Lintner et al. (2013), who found a minimum of the derivative for intermediate values of soil moisture. This is most likely a consequence of our formulation of advection which is here tailored to the case of a soil-moisture induced (or sea-breeze like) circulation.

The dependency of $\frac{\partial P_{\text{dry}}}{\partial \phi_{\text{dry}}}$ on $\eta_A, \eta_E$ is illustrated for 3 values of $\mathcal{B}$ in Fig. 9. Not surprisingly using a value of $\mathcal{B} = 1$ in Fig. 9 (b) gives a symmetric picture. Instead, different values of $\mathcal{B}$ slightly modify the conditions needed to obtain either a positive or negative derivative. Figure 9 overall shows that, as long as $\eta_A > \eta_E$ it is very unlikely to get a positive derivative, i.e. an increase of precipitation over the dry patch with increasing soil moisture. Only with values of $\mathcal{B}$ small enough, which would mean weaker and slower cold pools, the derivative may change sign even with $\eta_A > \eta_E$. Note how the values obtained in the `ID_` simulations (black point in Fig. 9 a) lead to a smaller value of the derivative, which is close to the zero-line. This agrees with the weaker sensitivity of precipitation to soil moisture observed in this case. Alternatively, evaporation should become more efficient of advection, i.e. $\eta_E \gg \eta_A$. This, however, does not happen in the performed simulations, which always lie in the rightmost lower quadrant (black points in Fig. 9).

These findings already answer the main question posed in the introduction and can be further generalized to the case when both $\phi_{\text{wet}}$ and $\phi_{\text{dry}}$ are changed at the same time. First of all, $\frac{\partial P_{\text{dry}}}{\partial \phi_{\text{wet}}}$ can be computed with the same method used before:

$$\frac{\partial P_{\text{dry}}}{\partial \phi_{\text{wet}}} = \eta_A \mathcal{B} \frac{\partial E_{\text{wet}}}{\partial \phi_{\text{wet}}} \tag{13}$$





given that the evaporation over the dry patch does not depend on the soil moisture of the wet patch. Second, the two derivatives $\frac{\partial P_{\mathrm{dry}}}{\partial \phi_{\mathrm{wet}}}, \frac{\partial P_{\mathrm{dry}}}{\partial \phi_{\mathrm{dry}}}$ can be combined to obtain the total precipitation change on the dry patch.

$$
\begin{aligned}
\Delta P_{\mathrm{dry}} &= \frac{\partial P_{\mathrm{dry}}}{\partial \phi_{\mathrm{dry}}} \Delta \phi_{\mathrm{dry}} + \frac{\partial P_{\mathrm{dry}}}{\partial \phi_{\mathrm{wet}}} \Delta \phi_{\mathrm{wet}} \\
&= (\eta_E - \eta_A \mathcal{B}) \frac{\partial E_{\mathrm{dry}}}{\partial \phi_{\mathrm{dry}}} \Delta \phi_{\mathrm{dry}} + \eta_A \mathcal{B} \frac{\partial E_{\mathrm{wet}}}{\partial \phi_{\mathrm{wet}}} \Delta \phi_{\mathrm{wet}}
\end{aligned}
\tag{14}
$$

Assuming that the soil type of both patches is the same from the various combinations between magnitude of $\phi_{\mathrm{wp}}$, $\phi_{\mathrm{dry,wet}}$ and $\phi_{\mathrm{crit}}$, only the case $\phi_{\mathrm{wp}} < \phi_{\mathrm{dry,wet}} < \phi_{\mathrm{crit}}$ is of interest. The other cases either revert to the previously discussed case (Eq. 12) or reduce to the trivial solution where only $\Delta \phi_{\mathrm{wet}}$ is affecting $\Delta P_{\mathrm{dry}}$. For $\phi_{\mathrm{wp}} < \phi_{\mathrm{dry,wet}} < \phi_{\mathrm{crit}}$ we obtain $\frac{\partial E_{\mathrm{dry}}}{\partial \phi_{\mathrm{dry}}} = \frac{\partial E_{\mathrm{wet}}}{\partial \phi_{\mathrm{wet}}}$. Thus, changes in precipitation in our idealized model can be formulated as

$$
\Delta P_{\mathrm{dry}} = \tau \frac{\mathcal{A} Q_{\mathrm{net}}}{\phi_{\mathrm{crit}} - \phi_{\mathrm{wp}}} \left( (\eta_E - \eta_A \mathcal{B}) \Delta \phi_{\mathrm{dry}} + \eta_A \mathcal{B} \Delta \phi_{\mathrm{wet}} \right)
\tag{15}
$$

The behaviour of Eq. 15 as function of $\Delta \phi_{\mathrm{dry}}$, $\Delta \phi_{\mathrm{wet}}$ and $\mathcal{B}$ can be investigated in Fig. 10. In the default configuration described in section 3.1 the soil moisture of the wet patch was kept constant, i.e. $\Delta \phi_{\mathrm{wet}} = 0$, while the soil moisture of the dry patch was increased, i.e. $\Delta \phi_{\mathrm{dry}} > 0$. Figure 10 (a) shows that, in the aforementioned case, $\Delta P_{\mathrm{dry}}$ is negative, as in our simulations. In this case decreasing $\phi_{\mathrm{dry}}$ and increasing $\phi_{\mathrm{wet}}$ is the most efficient way to increase precipitation.

Figure 10 (b) presents the case characteristic for the ID_ simulations. The flattening of the contour lines shows that there is little sensitivity on $\phi_{\mathrm{dry}}$, as already discussed previously. Mainly increasing $\phi_{\mathrm{wet}}$ would allow precipitation to increase. In the extreme case where $\mathcal{B}$ is further reduced (Fig. 10 c) the picture partly reverses. Both soil moisture of the wet and of the dry patch should be increased to sustain an increase of precipitation, as evaporation becomes now relevant and advection has a negligible contribution.

Figure 10 thus indicates that, in any case, the soil moisture of the wet patch should be increased to get more precipitation on the dry patch. The response to changes in soil moisture of the dry patch is more subtle, and the combination of the two responses can lead to positive or negative coupling depending on the atmosphere state. This may explain why in reality both signs of the coupling are observed with different atmospheric states.

## 5 Conclusions

Motivated by the ambiguous relationship between soil moisture, soil moisture heterogeneity and precipitation we designed idealized simulations of the convective diurnal cycle that make use of a coupled configuration of an atmospheric Large Eddy Simulation (LES) model and a land-surface model. The heterogeneity in the land surface was prescribed by dividing the domain into two patches with different initial values of soil moisture. Inspired by the results of the simulations, we specifically wanted to derive a simple conceptual model that retains the minimum parameters that control precipitation over a spatially drier patch. Moreover, we wanted to use such model to understand which is the most efficient way to increase precipitation by acting on soil moisture given the opposite control of soil moisture on advection and evaporation.





Since the main potential sources contributing to precipitation are constituted by remote moisture advection and local evapo-ration we first aim at disentangling the effects of these two on precipitation. Results from the simulations show, as expected, that the moisture advection over the dry patch decreases with increasing local soil moisture, while evaporation increases. The interplay between these two effects produces a decrease of precipitation with increasing values of local soil moisture for the

considered case.

More importantly the simulation results indicated that such a decrease can only be correctly reproduced by assuming that ad-vection and evaporation processes contribute differently to precipitation. Hence we model precipitation as the sum of advection and evaporation each weighted by its own efficiency (see Eq. 2). By using two efficiencies they become independent of soil moisture and only depend on the initial atmospheric state.

As a second step we conceptualize the variations of evaporation and advection with soil moisture. Evaporation can be approx-imated using the *bucket* model owing to Budyko (1961) (see Eq. 3). The advection is estimate as the product of the breeze front, triggered by the surface heterogeneity, velocity and the gradient in near-surface specific humidity (see Eq. 5). The latter depends on the gradient in latent heat fluxes, whereas the former is surprisingly independent of the degree of soil moisture het-erogeneity. This is so because the breeze front velocity is, after a certain time, fully determined by cold pools whose strength

is mainly controlled by the upper tropospheric profile.

Putting all the results together indicates that the variations of precipitation over the dry patch do not depend on the actual soil moisture value. This is due to the fact that the derivation of the functional forms of advection and evaporation ends up to be very similar and cancels out the dependency on soil moisture. The parameters that control the variations of precipitation with local soil moisture are the aforementioned efficiencies and a scale parameter that defines the magnitude of the advection. All

these parameters depend solely on the atmospheric state. According to the values of these parameters as estimated from the simulations the most efficient way to increase precipitation over the dry patch is always to decrease soil moisture over the dry patch. Thus, one can say that, in order to have more precipitation over spatially drier areas, more precipitation should first fall on spatially wetter ones. In other words, the most efficient way to obtain more precipitation over dry areas is to let them dry out for a long time so that a stronger gradient can build up and thus produce more explosive convective events due to a stronger

meso-scale circulation.

However, if either the efficiency of evaporation becomes much larger than the one of advection or the scale of advection de-creases under a certain threshold then the response of precipitation can be reversed. Although we did not find any evidence of this behaviour for the two atmospheric profiles tested in this work it would be interesting as a next step to derive these three parameters predicted by the conceptual model from more realistic simulations to infer the frequency of occurrence of the

various precipitation regimes.





## Appendix A: Computation of the advection as residual term

The advection of every tracer spatially averaged over a certain area $A_{\mathrm{area}}$ is computed directly from the moisture balance equation as a residual. We use the following formulation:

$$\frac{1}{\rho_w}\int\limits_0^\tau\int\limits_0^H \frac{\mathrm{D}}{\mathrm{D}t}\left(\rho_a q_{\mathrm{tot}}\right)\Big|_{\mathrm{area}}\mathrm{d}z\,\mathrm{d}t = \frac{1}{\rho_w}\int\limits_0^\tau \frac{\mathrm{D}\overline{q}_{\mathrm{tot}}}{\mathrm{D}t}\Big|_{\mathrm{area}}\mathrm{d}t = E_{\mathrm{area}} - P_{\mathrm{area}} \tag{A1}$$

where D indicates the total derivative, $P_{\mathrm{area}}$ is the area-averaged accumulated precipitation, $E_{\mathrm{area}}$ the area-averaged accumulated evaporation, $\overline{q}_{\mathrm{tot}}$ is the area-averaged vertically integrated sum of all tracers (water vapour, clouds, rain, snow, ice, graupel and hail) mixing ratios, $\rho_w$ the density of water and $\rho_a$ the air density. As the quantities on the right-hand-side are accumulated over the length of the diurnal cycle $\tau$, the left-hand-side is explicitly integrated over time. $H$ indicates the top of the simulation domain. The total derivative term can be further divided into its advective term:

$$A_{\mathrm{area}} = -\frac{1}{\rho_w}\int\limits_0^\tau \nabla(\overline{q}_{\mathrm{tot}} \cdot \mathbf{v})\,\mathrm{d}t \tag{A2}$$

and a local derivative:

$$\frac{1}{\rho_w}\int\limits_0^\tau \frac{\partial\overline{q}_{\mathrm{tot}}}{\partial t}\Big|_{\mathrm{area}}\mathrm{d}t = A_{\mathrm{area}} + E_{\mathrm{area}} - P_{\mathrm{area}} \tag{A3}$$

Although other studies only considered the advection of water vapour, i.e. of $\overline{q}_v$, in order to close the balance it is necessary to consider all species. In fact, although $\overline{q}_i, \overline{q}_g, \overline{q}_h, \overline{q}_s$ are order of magnitudes smaller than $\overline{q}_v, \overline{q}_c, \overline{q}_r$, their variations over time are

not, so that neglecting these terms in Eq. A3 would lead to an unbalance.

From the 5-min simulation output we use Eq. A3 and estimate the advection as the residual $R_{\mathrm{area}} = \int_0^\tau \frac{\partial\overline{q}_{\mathrm{tot}}}{\partial t}\Big|_{\mathrm{area}}\mathrm{d}t + P_{\mathrm{area}} - E_{\mathrm{area}} \equiv A_{\mathrm{area}}$. We verify that, when averaged over the entire domain, $R_{\mathrm{dom}} = 0$.

*Competing interests.* No competing interests are present.

*Acknowledgements.* This research was supported by the Hans Ertel Center for Weather Research (HErZ), a collaborative project involving
universities across Germany, the Deutscher WetterDienst (DWD) and funded by the Federal Ministry of Transport and Digital Infrastructure (BMVI). The simulations were performed using the facilities of the Detusches KlimaRechenZentrum (DKRZ) and in particular the new supercomputer *Mistral*.



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



**Figure 3.** Tracking of the breeze front for the case DA_20_100. (a) Hovmöller diagram (distance from domain center vs. time) of the $y$-averaged zonal wind at a height of approximately 100 m above the surface. Dots indicate the position of the front tracked every 15 minutes (see text for details). (b) Front inland propagation (black line) with respect to the center of the domain [km] and front speed (red line) derived using finite differences [m s$^{-1}$]



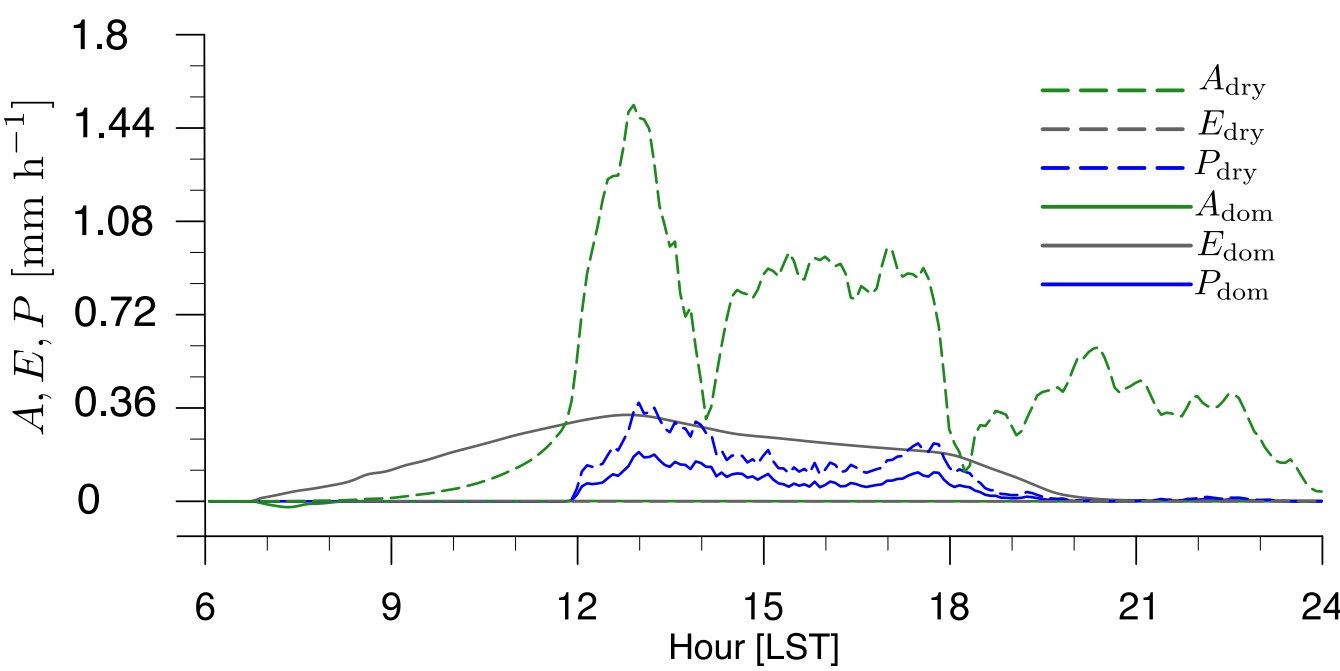

**Figure 4.** Different terms of the moisture balance (Eq. A3) computed for the entire domain (subscript dom, solid lines) and for the dry patch (subscript dry, dashed lines) in the `DA_20_100` case. $A$ indicates advection, $E$ evaporation and $P$ precipitation. Units are mm h$^{-1}$. Note that all variables in this figure are instantaneous. See Appendix A for a comprehensive explanation of all the symbols.





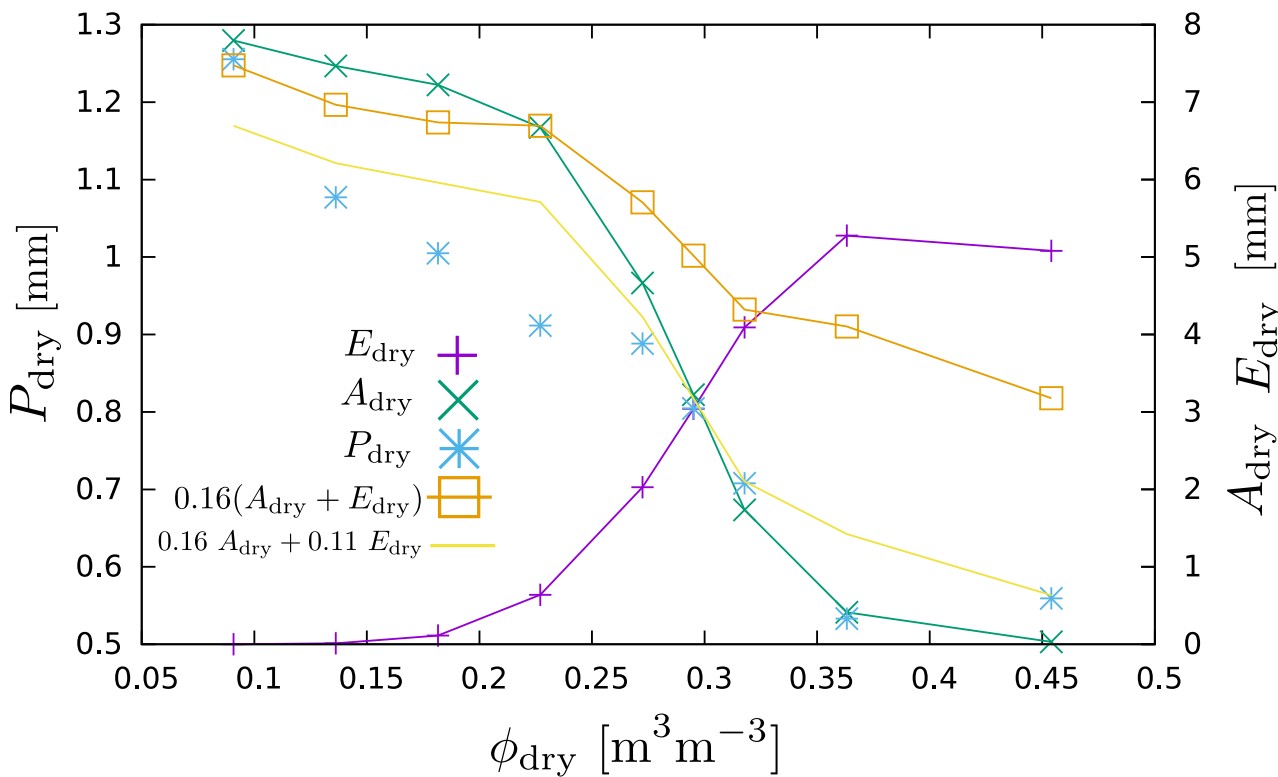

**Figure 5.** Values of advection (green line and crosses), evaporation (purple line and crosses) and precipitation (blue asterisks) of Tab. 2 as function of soil moisture. The orange line represents an estimate of precipitation obtained as weighted sum of advection and evaporation, i.e. $0.16(A_{dry} + E_{dry})$ while the yellow line represents a similar estimate used by obtaining two different efficiencies, i.e. $0.16\,A_{dry} + 0.11\,E_{dry}$.





**Figure 6.** Hovmöller diagram of precipitation rate [mm h$^{-1}$] in case ID_20_100 and DA_20_100.





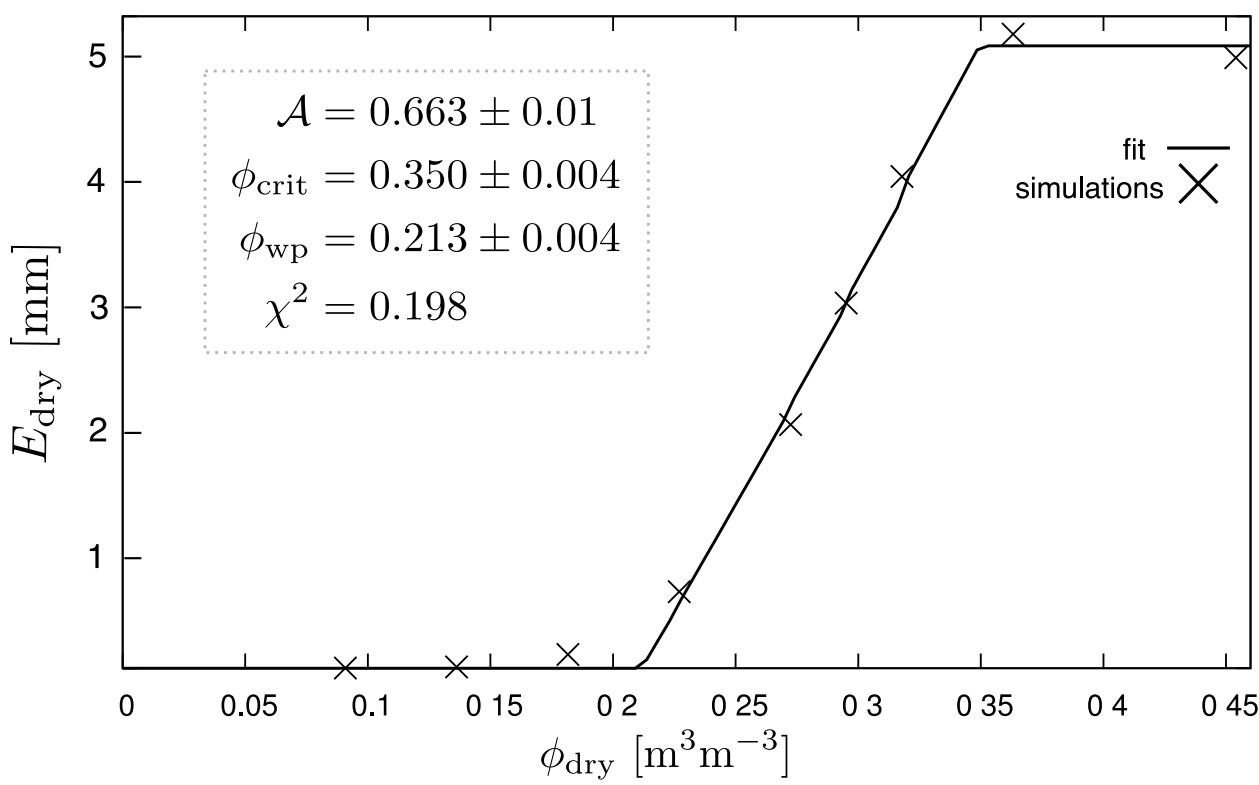

**Figure 7.** Fit of $E_{\mathrm{dry}}$ with values obtained from the simulations of the default configuration (`DA_20_100` to `DA_100_100`). Crosses indicate values obtained from simulations while line indicates the fit performed using the function of Eq. 4. Upper left inset show the values obtained by the fit together with absolute errors and the $\chi^2$ value.



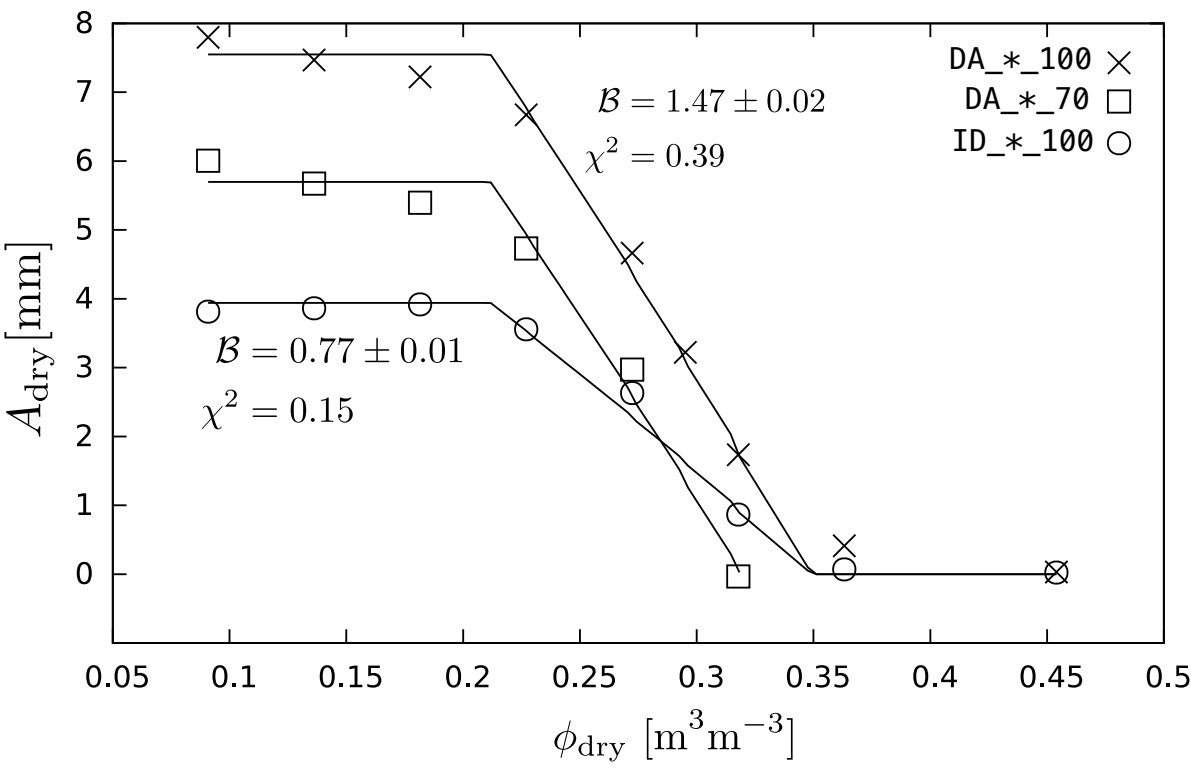

**Figure 8.** Fit of the advection in cases `DA_*_100`, `DA_*_70` and `ID_*_100`. Symbols indicate the values obtained from the simulations while lines represent the fit performed using Eq. 9. The different values of $\mathcal{B}$ obtained are reported in the insets, together with the absolute error and the $\chi^2$ value.





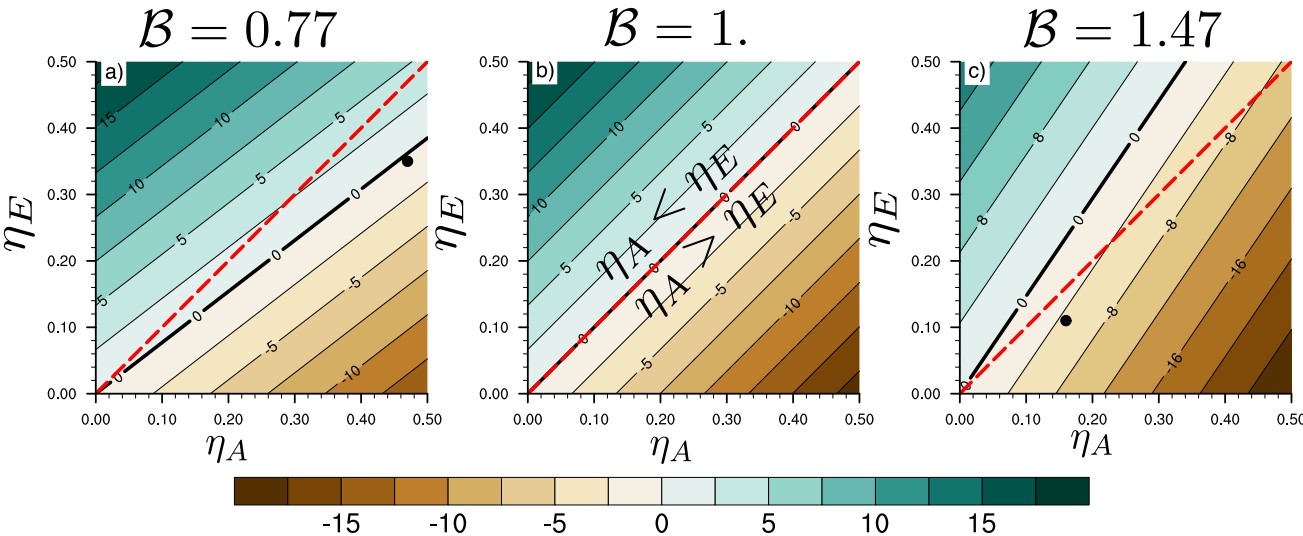

**Figure 9.** Contour plot of $\frac{\partial P_{\mathrm{dry}}}{\phi_{\mathrm{dry}}}$ [mm m$^3$ m$^{-3}$] as a function of $\eta_A, \eta_E$ for different values of the parameter $\mathcal{B}$. The black points in (a) and (c) are placed using the efficiencies obtained in the `ID_` and `DA_` cases, respectively. The dashed red line distinguishes the areas where $\eta_A > \eta_E$ and vice versa. Note the symmetric color scale and the thicker zero contour line.





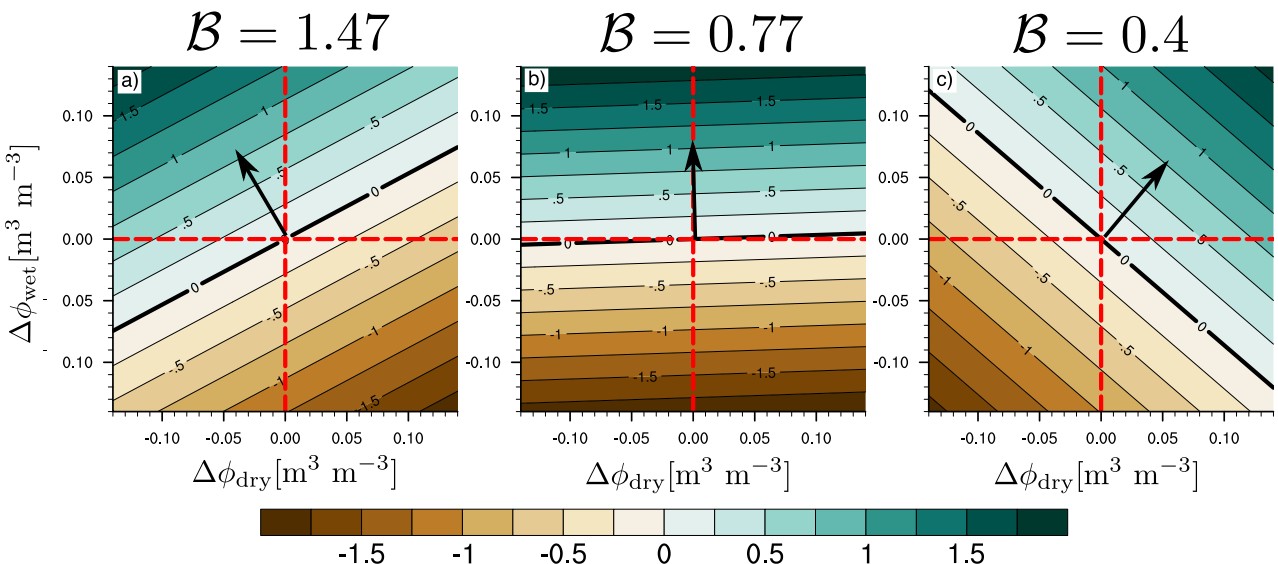

**Figure 10.** $\Delta P_{\mathrm{dry}}$ as function of $\Delta\phi_{\mathrm{dry}}$ and $\Delta\phi_{\mathrm{wet}}$ for different values of the parameter $\mathcal{B}$. The $x$ and $y$ axes represent the variation of $\phi_{\mathrm{dry}}$ and $\phi_{\mathrm{wet}}$, respectively. Note that the maximum variation is $\phi_{\mathrm{crit}} - \phi_{\mathrm{wp}}$, as $\Delta P_{\mathrm{dry}}$ is computed for the regime $\phi_{\mathrm{wp}} < \phi_{\mathrm{dry,wet}} < \phi_{\mathrm{crit}}$. The red dashed lines indicate no variation of the soil moisture of either one of the patches. The efficiencies are set to $(\eta_A, \eta_E) = (0.16, 0.11)$ in (a) and to $(\eta_A, \eta_E) = (0.47, 0.35)$ in (b) and (c) to match the simulation results. The black arrow indicates the direction of maximum growth, i.e. when an increase of precipitation is expected.