# Peer review of "A simplified model of precipitation enhancement over a heterogeneous surface"

_Hydrology and Earth System Sciences, 2017_

## Referee Comment (RC1) · Anonymous Referee #1 · 12 Oct 2017

General Comments:

This paper uses an idealized simulation with prescribed soil moisture gradients to derive a simplified algorithm that represents the amount of precipitation generated by local evaporation and advection terms. The authors note that previous studies have qualitatively shown how soil moisture gradients and atmospheric profile influence precipitation, and state that their goal is to quantitatively isolate the primary drivers of precipitation. I believe their methods, i.e. using an idealized model with prescribed soil moisture gradients, are sound, and their results are relevant.

Overall, I find that the paper convincingly demonstrates the relative important of soil moisture gradients over the absolute magnitude of soil moisture, which makes sense physically, but it glosses over some other important points that deserve more expla-

nation, such as the importance of the atmospheric profiles. Also, the derivation of the algorithm they use seems fine, but needs some clarification in order for the reader to be able to completely recreate their results.

The second stated goal of the paper is to determine "what is the relative role of the atmosphere, or in other words the efficiency in converting these potential moisture sources into precipitation." Terms that represent the efficiency of advection and evaporation are derived, but there is no discussion of how the actual atmospheric profile impacts those terms, which then detracts from the significance of these findings. Also, while the authors cite publications that use the two atmospheric profiles utilized in the model simulations, they do not display them in a figure or discuss them in any way. This leaves the reader wondering what the difference is between them, what the profiles are like, and how these profiles could affect the results. For example, a profile that is more unstable could increase convection and strengthen the circulation, however there is no context like this provided in the paper. Also, I looked up the two profiles in the cited publications and found it difficult to compare them because they are presented in different formats. Because of these oversights, the reader is left unsure why the authors included two different profiles in the first place, and how the atmospheric profile impacts the authors' findings.

Specific Comments:

1. page 3 line 10-12: "the change of precipitation with soil moisture does not depend on the soil moisture content itself and that the most efficient way to increase precipitation consists in increasing the surface wetness gradient.", but page 1 line 8-9: "these changes surprisingly do not depend on soil moisture itself but instead purely on parameters that describe the atmospheric initial state." — is it the atmospheric state or the soil moisture gradient that is most important? Also, see my other comments about the importance of addressing the atmospheric state more thoroughly in the paper.

2. Page 10, Line 11: "In order to test the validity of the theory proposed in section 2"

is confusing. This is stated in section 2, and I'm not sure what the theory is. Suggest repeating what the theory is or otherwise clarifying here.

3. Page 4: Please clarify why the "dry-soil advantage profile of Findell and Eltahir 2003" is used and why it is appropriate for this investigation.

4. Please include an additional figure with the two atmospheric profiles (from Findell and Eltahir 2003 and Schlemmer et al. 2012).

5. Page 4: Please clarify why the Schlemmer et al. (2012) profile is used over a different one, what question is answered by including it in the study, and how it differs from the profile from Findell and Eltahir 2003.

6. Figure 2: This figure takes some time and effort to interpret. It would be easier for the reader if vectors were used in place of windspeed contours and if the "dry" and "wet" sides are labeled. Also, please add a sentence to the text explicitly stating which side in Figure 2 is warmer (and why) and which direction the front is propagating. This all may seem obvious, and is stated more explicitly later in the text, but to the first-time reader it takes time to put it all together while examining figure 2.

7. Page 8 Line 24: Clarify what "the fact that one efficiency doesn't match well" means. Which efficiency? And it doesn't match well with what?

8. Page 8 Line 29: which sounding is "another sounding?". Also see previous comments about soundings. This would be a good place to spend some time discussing what it is about the two profiles that result in efficiencies that are higher than with the first sounding.

9. Page 9 Line 2: "a weaker sensitivity of that particular atmospheric state" ….see above comments about the atmospheric profiles. This reference is too vague, and needs more explanation.

10. Page 10 Lines 18-19. This is the first point that the soil type is referenced. The data and methods should include a sentence stating the soil type used in the simulation, the

reason why it is used, and its field capacity.

11. Page 12: The derivation of beta needs some more explanation. Was it derived using a best fit method from Figure 3? I'm not sure.

12. As a reader, it was difficult to get through sections 4.2 and 4.3. There were some jumps in the logic between equations that were hard to follow, and not all terms were defined (see above). I think if the authors revisit these sections and provide more explicit explanations even where they think the transitions should be obvious, it will help the reader finish the paper.

13. Page 15, Line 20: "these parameters depend solely on the atmospheric state." See above comments.

14. Figures 9 and 10: These are important figures. More explanation of these figures is needed, particularly the significance of $n_a < n_b$ (and visa versa) and of beta, and what that means physically. As a reader, I found myself quite bogged down by this point and it was difficult to extract what the authors were hoping to convey with these figures.

Technical Corrections:

Page 1 Line 1-2: For clarity, I suggest rewording the first sentence of the abstract to read "Soil moisture heterogeneities influence the onset of convection and subsequent evolution of thunderstorms producing heavy precipitation through the triggering of mesoscale circulations."

Page 1 Line 6: Suggest rewording to read "A key element of the model is the representation of precipitation as a weighted sum"

Page 1 Line 18: Suggest rewording to read "and which can then affect the distribution of precipitation."

Page 2 Line 17: Please clarify what is meant by "a negative spatial coupling coexists

together with a positive temporal coupling."

Page 3 Section 2.1 heading: Is the subheading "2.1 Experimental Design" needed here? There are no other subsections in Section 2.

Page 3 Line 3: "overt" should be "over"

Page 6 Lines 6-8: This sentence is difficult to understand. I suggest rewording it.

Page 7 Line 11: "It is immediate to verify" is awkward. I suggest rewording.

Page 8 Line 5: "firstly" should be "first"

Page 8 Line 23: The text states $n_a = 0.15$ and $n_b = 0.10$, but Figure 5 states that they are 0.16 and 0.11, respectively.

Page 10 Line 19: what is "the expected one"? Please clarify.

Equation 6: I couldn't find a definition for $L_{front}$ anywhere in the text. Please include a definition here.

---

## Referee Comment (RC2) · Anonymous Referee #2 · 21 Nov 2017

Review of manuscript (hess-2017-547): A simplified model of precipitation enhancement over a heterogeneous surface By: G. Cioni and C. Hohenegger Recommendation: Major Revision Anonymous: Yes

General Comments This study aims to identify the most important parameters that impact rainfall variations over spatially drier patches. Relying on idealized simulations and a simplified model, the authors concluded that precipitation changes over a heterogeneous surface do not depend on soil moisture, but the initial atmospheric state. The research is interesting. However, the manuscript needs to be substantially clarified and the formulation of the simplified model should be further justified.

Major Comments 1. The simulations As described in Section 2, the simulations were conducted using an atmospheric model (ICON-LEM) coupled with a land surface model

(TERRA-ML). Accordingly, it appears that soil moisture over both the dry and wet patches evolves as the model integrates forward and soil moisture in each experiment is only specified at the initial time. I am not completely sure about whether this is the case, because assumptions in the simplified model are more consistent with simulations using constant soil moisture values throughout the model run. Please clarify.

Also, please briefly describe the purpose of reducing dynamic contributions of advection on precipitation when setting up the size of the simulation domain (Pg. 3, lines 26-28).

2. The simplified model 1) Assumptions According to Section 4, the authors assumed that "Ewet does not depend on ÏŢdry" (Pg. 13, line 1) and "evaporation over the dry patch does not depend on the soil moisture of the wet patch" (Pg. 14, line 1). These two assumptions are needed to get the key results (Eqs. 10, and 15), but not clearly justified. When either ÏŢdry or ÏŢwet varies, should not precipitation over the wet or dry patches change, which in turn impact Ewet or Edry through the impact on soil moisture therein?

On Pg. 11 (lines 2-7), it is assumed that the advection of water vapor and hydrometeors is mainly constrained in the boundary layer. As shown in Fig. 2, however, the return flow at ∼1-3 km is not negligible. Could you please justify this assumption further?

2) Derivations Please provide more details on how to approximate Eq. 5 to get Eq. 6, and how Eq. 7 is obtained.

To get Eq. 8, it seems that one has to assume the vertical extent of moistening process due to latent heat flux, Hmoist, is the same over the dry and wet patches. Is Hmoist related to turbulent eddies? If so, this assumption can be problematic because low-level temperature differences are up to ∼4 K between the dry and wet patches (Fig. 2), where sensible heat flux differences can reach 280 W/m2 (Pg. 5, lines 5-6).

3) Comparison to Lintner et al. (2013) As noted in the article (Pg. 10, lines 26-30),

feedbacks between the land-surface and atmosphere are neglected in the simplified model after taking evaporation as Eq. 4. Consequently, it is not unexpected that soil moisture can be irrelevant to precipitation change in the simplified model. If the formulation of evaporation in Lintner et al. (2013), where land-atmosphere interactions are considered, is used in the derivation, will the theoretical model proposed here still be valid?

4) Precipitation efficiency associated with evaporation and advection Could you please elaborate further on why precipitation efficiency is independent of soil moisture? Although the authors showed that precipitation efficiency associated with advection is independent of evaporation using the extreme case DA_20_100, where Edry is negligible, it is not clear on why precipitation efficiency associated with evaporation is independent of soil moisture.

Overall, it is hard to evaluate the simplified model according to how it is presented. The conclusion that precipitation change over a heterogenous surface is independent of soil moisture can be an artifact that land-atmosphere interactions are eliminated in the theoretical model.

2. Writing The manuscript requires an editorial revision to correct wording issues. Some sentences are either awkward or redundant. For example, "..., in a nutshell, ..." (Pg. 6, line 4), ".. thanks to the previous section ..." (Pg. 12, line 12) and etc. can be removed.

Minor Comments 1. Are equations 5 and A2 written correctly as advection? It is also unusual to have dot product between a scalar (qtot) and a vector (ufront or v).

2. Pg. 2, lines 16-17: Please clarify further on Guillod et al. (2015), what does "... a negative spatial coupling coexists together with a positive temporal coupling" mean and indicate?

3. Pg. 3, line 25: Why a rectangular domain can limit computational cost?

4. Pg. 9, lines 6-7: Please provide relevant evidences on ". . . several secondary events develop due to the waves propagating away from the collision".

5. Fig. 2: It can be better to show wind as vectors, rather than contours.

6. Fig. 9: Change "$\partial Pdry/\ddot{I}\c{T}dry$" as "$\partial Pdry/\partial\ddot{I}\c{T}dry$".

7. Table 1: Change "Name" as "Experiment".

---

## Author Comment (AC1) · 18 Dec 2017

**General Comments:**

This paper uses an idealized simulation with prescribed soil moisture gradients to derive a simplified algorithm that represents the amount of precipitation generated by local evaporation and advection terms. The authors note that previous studies have qualitatively shown how soil moisture gradients and atmospheric profile influence precipitation, and state that their goal is to quantitatively isolate the primary drivers of precipitation. I believe their methods, i.e. using an idealized model with prescribed soil moisture gradients, are sound, and their results are relevant. Overall, I find that the paper convincingly demonstrates the relative important of soil moisture gradients over the absolute magnitude of soil moisture, which makes sense physically, but it glosses over some other important points that deserve more explanation, such as the importance of the atmospheric profiles. Also, the derivation of the algorithm they use seems fine, but needs some clarification in order for the reader to be able to completely recreate their results. The second stated goal of the paper is to determine "what is the relative role of the atmosphere, or in other words the efficiency in converting these potential moisture sources into precipitation." Terms that represent the efficiency of advection and evaporation are derived, but there is no discussion of how the actual atmospheric profile impacts those terms, which then detracts from the significance of these findings. Also, while the authors cite publications that use the two atmospheric profiles utilized in the model simulations, they do not display them in a figure or discuss them in any way. This leaves the reader wondering what the difference is between them, what the profiles are like, and how these profiles could affect the results. For example, a profile that is more unstable could increase convection and strengthen the circulation, however there is no context like this provided in the paper. Also, I looked up the two profiles in the cited publications and found it difficult to compare them because they are presented in different formats. Because of these oversights, the reader is left unsure why the authors included two different profiles in the first place, and how the atmospheric profile impacts the authors' findings.

**Response**:

We thank the reviewer for his/her comments which helped us to revise the parts of the manuscript which were not clear. As suggested by the reviewer we clarified why the particular soundings employed in the study were chosen and decided to add a figure with two skew-t diagrams relative to the different atmospheric soundings. Furthermore, we elaborated more on the physical meaning of the efficiencies and why the atmospheric profiles have different ones. In the following we present the responses to the reviewer specific comments and technical corrections, which also answer the questions presented in the general comments.

**Specific Comments:**

1. page 3 line 10-12: "the change of precipitation with soil moisture does not depend on the soil moisture content itself and that the most efficient way to increase precipitation consists in increasing the surface wetness gradient.", but page 1 line 8-9: "these changes surprisingly do not depend on soil moisture itself but instead purely on parameters that describe the atmospheric initial state." — is it the atmospheric state or the soil moisture gradient that is most important? Also, see my other comments about the importance of addressing the atmospheric state more thoroughly in the paper.

**Response**: we agree with the reviewer that the presence of both sentences was misleading. For this reason we modified them in the manuscript and added some clarification notes on the dependency of precipitation on both soil moisture and the atmospheric state. We revised section 4.3 and stressed the conclusion that, although the derivative of precipitation does not depend on soil moisture but just on the atmospheric state through the efficiencies and the B parameter, the absolute value of precipitation does depend on soil moisture as shown in Eq. 15.

2. Page 10, Line 11: "In order to test the validity of the theory proposed in section 2" is confusing. This is stated in section 2, and I'm not sure what the theory is. Suggest repeating what the theory is or otherwise clarifying here.

**Response**: We apologise for the wrong reference: it should have been Section 3 instead. We corrected this in the manuscript.

3. Page 4: Please clarify why the "dry-soil advantage profile of Findell and Eltahir 2003" is used and why it is appropriate for this investigation.
**Response**: This particular sounding was observed on 23 July 1999 in Lincoln, Illinois (USA) and was chosen as a typical example by Findell and Eltahir 2003 for cases when a strong heating of the surface forces the triggering of convection. Given that on the dry patch low soil moisture availability causes strong sensible heat fluxes to heat the air above we though that using this sounding would produce the strongest response in the atmosphere.

4. Please include an additional figure with the two atmospheric profiles (from Findell and Eltahir 2003 and Schlemmer et al. 2012).
**Response**: we added an additional figure in the manuscript with a skew-t diagram of the two soundings (see Fig.1 of this document).

[Figure]

**Figure 1:** Skew-t diagrams for the sounding used in the simulations.

5. Page 4: Please clarify why the Schlemmer et al. (2012) profile is used over a different one, what question is answered by including it in the study, and how it differs from the profile from Findell and Eltahir 2003.
**Response**: we used the profile of Schlemmer et al. (2012) for one main reasons, namely that it greatly differs from the sounding of Findell and Eltahir (2003). In particular it has a lower surface temperature and lower integrated water vapour content although having a larger initial instability. This allows us to test the idealized model and show that efficiencies and the B parameter do depend on the atmospheric state. We performed additional simulations using the second sounding presented in Findell and Eltahir 2003 but, as the results were similar to the DA case, we didn't include those in the manuscript. We added a few sentences in the manuscript to justify our choice.

6. Figure 2: This figure takes some time and effort to interpret. It would be easier for the reader if vectors were used in place of windspeed contours and if the "dry" and "wet" sides are labeled. Also, please add a sentence to the text explicitly stating which side in Figure 2 is warmer (and why) and which direction the front is propagating. This all may seem obvious, and is stated more explicitly later in the text, but to the first-time reader it takes time to put it all together while examining figure 2.
**Response**: We changed Fig. 2 in order to ease its interpretation. We used vectors instead of contours to indicate zonal winds and used explicit labels for the dry and wet patches. We think now it is clear

where the patches are located so that there is no need to add another sentence to say which side of Figure 2 is warmer and in which direction the front is propagating.

7. Page 8 Line 24: Clarify what "the fact that one efficiency doesn't match well" means. Which efficiency? And it doesn't match well with what?
**Response**: We meant that one efficiency is not enough to describe the variations of precipitation. As shown in Fig. 6 when using a single efficiency the decrease of precipitation with increasing value of soil moisture on the dry patch cannot be captured. We rephrase this sentence in the manuscript.

8. Page 8 Line 29: which sounding is "another sounding?". Also see previous comments about soundings. This would be a good place to spend some time discussing what it is about the two profiles that result in efficiencies that are higher than with the first sounding.
**Response**: We were referring to the sounding of Schlemmer et al. (2012) so we added this explicit reference in the text. We think that the higher efficiencies obtained with the Schlemmer et al. (2012) sounding are due to a combination of different effects. One of those is the different convection triggering. With the sounding of Schlemmer et al. (2012) convection is triggered almost 1 hour before than with the sounding of Findell and Eltahir (2003). This allows the atmosphere to fully exploit the instability caused by the morning heating and to develop a stronger front propagation, as shown in Fig. 6 of the manuscript. Although the maximum advection of moisture over the dry patch in the ID cases is smaller than the one of the DA cases, the atmosphere is able to efficiently convert it into precipitation, thus leading to larger efficiencies. This is also evident in Fig. 2 of this

[Figure]

**Figure 2:** meridional average of convective available potential energy (CAPE, J/kg) for the simulation with the ID sounding (upper panel) and the DA sounding (lower panel). For both simulations the extreme case with a dry patch at 20% saturation is used. The lines indicate the value at different hours during the day (6 LST, initialization time, to 15 LST).

document where the value of CAPE is plotted as a function of time and x-dimension only. When using the DA sounding it can be seen that convective potential instability at 15 LST is larger than the one at the initial time over both patches, while in the ID sounding the opposite happens. Furthermore, the Findell and Eltahir (2003) sounding was not prone to the development of intense rain events, as shown in Cioni & Hohenegger (2017), thus the smaller efficiencies. We added this brief discussion to the manuscript.

9. Page 9 Line 2: "a weaker sensitivity of that particular atmospheric state" . . ..see above comments about the atmospheric profiles. This reference is too vague, and needs more explanation.
**Response**: We meant that with this sounding precipitation amounts seem not to depend much on the advected moisture, as the estimated B parameter is smaller. We corrected the reference on the manuscript.

10. Page 10 Lines 18-19. This is the first point that the soil type is referenced. The data and methods should include a sentence stating the soil type used in the simulation, the reason why it is used, and its field capacity.
**Response**: We added a sentence stating the soil type used (5, loam) and the reason why it was chosen, specifically because it is the most frequent soil type over Germany. Also the field capacity and the wilting point were added to the manuscript.

11. Page 12: The derivation of beta needs some more explanation. Was it derived using a best fit method from Figure 3? I'm not sure.
**Response**: as explained in Lines 11-20 the parameter B is obtained through a best fit of the values of advection and evaporation, where evaporation is approximated through Eq. 4. We further clarified this aspect in the manuscript.

12. As a reader, it was difficult to get through sections 4.2 and 4.3. There were some jumps in the logic between equations that were hard to follow, and not all terms were defined (see above). I think if the authors revisit these sections and provide more explicit explanations even where they think the transitions should be obvious, it will help the reader finish the paper.
**Response**: We agree with the reviewer. For this reason we revised section 4.2 and 4.3 by expanding all the steps used in deriving Eq. 8 and 9 from Eq. 5 and Eq. 15. We added a definition for all the missing variables and further expanded the explanations in the text.

13. Page 15, Line 20: "these parameters depend solely on the atmospheric state." See above comments.
**Response**: see answers above.

14. Figures 9 and 10: These are important figures. More explanation of these figures is needed, particularly the significance of $n_a < n_b$ (and visa versa) and of beta, and what that means physically. As a reader, I found myself quite bogged down by this point and it was difficult to extract what the authors were hoping to convey with these figures.
**Response**: We agree with the reviewer: these are the most important figures of the paper and deserve more explanation. We think that the significance of $n_a < n_e$ (and vice-versa) is related to the different way advection and evaporation sources are used by the atmosphere to produce precipitation. In our simulations we always find that the efficiency of advection is larger than the efficiency of evaporation which would mean that the atmosphere is somehow able to use more of the advected moisture than of the evaporated one to produce precipitation. The B parameter, instead, appears to be an additional parameter which describes the importance of advection. We think that this is related to the strength of cold pools, which enhance the advection processes. We revised the explanation of these figures.

**Technical Corrections:**

Page 1 Line 1-2: For clarity, I suggest rewording the first sentence of the abstract to read "Soil moisture heterogeneities influence the onset of convection and subsequent evolution of thunderstorms producing heavy precipitation through the triggering of mesoscale circulations."
**Response**: Corrected.

Page 1 Line 6: Suggest rewording to read "A key element of the model is the representation of precipitation as a weighted sum"
**Response**: Corrected.

Page 1 Line 18: Suggest rewording to read "and which can then affect the distribution of precipitation."
**Response**: corrected

Page 2 Line 17: Please clarify what is meant by "a negative spatial coupling coexists together with a positive temporal coupling."
**Response**: We added a clarification sentence: " That is, areas drier than their surrounding (spatial component) but wetter than the climatological value (temporal component) may receive more precipitation than other ones"

Page 3 Section 2.1 heading: Is the subheading "2.1 Experimental Design" needed here? There are no other subsections in Section 2.
**Response**: corrected.

Page 3 Line 3: "overt" should be "over"
**Response**: corrected.

Page 6 Lines 6-8: This sentence is difficult to understand. I suggest rewording it.
**Response**: We added further clarification when describing the algorithm: "More specifically, at the first two time instants the maximum is searched over the entire dry patch while from the third time step onward the maximum search is performed in a box centered on a first guess obtained from a simple linear extrapolation of the previous time instants. "

Page 7 Line 11: "It is immediate to verify" is awkward. I suggest rewording.
**Response**: we changed the sentence to "it can be verified".

Page 8 Line 5: "firstly" should be "first"
**Response**: corrected.

Page 8 Line 23: The text states $n\_a = 0.15$ and $n\_b = 0.10$, but Figure 5 states that they are 0.16 and 0.11, respectively.
**Response**: This was not a typo and deserve some explanation. Both efficiencies can be estimated either from the extreme case DA_20_100 and DA_100_100, assuming that evaporation or advection, respectively, are negligible, or from fitting Eq. 2 to the value obtained in the simulations (as done in Fig.5). The estimate of the efficiencies obtained with these two different methods are just slightly different (0.15 vs. 0.16 and 0.10 vs. 0.11), thus the confusion. We clarified this aspect in the manuscript to make clear that one could use both methods to estimate the efficiencies.

Page 10 Line 19: what is "the expected one"? Please clarify.
**Response**: we meant the wilting point of the particular soil employed in the simulations. This is now clarified in the manuscript, also in light of the previous comment about the soil characteristics.

Equation 6: I couldn't find a definition for L_front anywhere in the text. Please include a definition here.

**Response**: L_front represents the penetration lenght of the front. We added a definition for L_front in section 4.2.

---

## Author Comment (AC2) · 18 Dec 2017

**General Comments**
This study aims to identify the most important parameters that impact rainfall variations over spatially drier patches. Relying on idealized simulations and a simplified model, the authors concluded that precipitation changes over a heterogeneous surface do not depend on soil moisture, but the initial atmospheric state. The research is interesting. However, the manuscript needs to be substantially clarified and the formulation of the simplified model should be further justified.
**Response**: We thank the reviewer for his/her precious comments which helped us to revise many sections of the manuscript which were not clear. In the following we will answer all the major and minor comments of the reviewer.

**Major Comments**
**1. The simulations**
As described in Section 2, the simulations were conducted using an atmospheric model (ICON-LEM) coupled with a land surface model (TERRA-ML). Accordingly, it appears that soil moisture over both the dry and wet patches evolves as the model integrates forward and soil moisture in each experiment is only specified at the initial time. I am not completely sure about whether this is the case, because assumptions in the simplified model are more consistent with simulations using constant soil moisture values throughout the model run. Please clarify. Also, please briefly describe the purpose of reducing dynamic contributions of advection on precipitation when setting up the size of the simulation domain (Pg. 3, lines 26-28).
**Response**: The referee is right: soil moisture in our setup is prescribed at the initial time and then freely evolves during the day as a response to the atmospheric forcing (precipitation, evaporation...) and to the soil model.  Thus, whereas in the formulation of evaporation of the simplified model (Eq. 4) we used the value of soil moisture at initialisation time and assumes it stays constant, soil moisture, and thus evaporation, will change in the simulations. However changes in soil moisture over one diurnal cycle are not expected to be so strong to significantly feed back on evaporation and precipitation. We investigate this by computing the daily average value of soil moisture in the simulations and comparing it to its initial value (see Tab. 1 of this document). Except in the

| Case | Initial soil moisture (dry patch average) [m³ m⁻³] | Diurnally-averaged soil moisture (dry patch average) [m³ m⁻³] |
|---|---|---|
| **DA_20_100** | 0.0908 | 0.109 |
| **DA_50_100** | 0.227 | 0.227 |
| **DA_65_100** | 0.2951 | 0.270 |
| **DA_100_100** | 0.454 | 0.291 |

**Table 1**: Prescribed initial value of volumetric soil moisture and diurnally-averaged value over the entire period of the simulation for different cases (first column). All values are considered averaged over the dry patch, although for the initial soil moisture we prescribe the same value everywhere over the dry patch (see manuscript).

DA_100_100 case, the values are fairly similar. The big difference between initial soil moisture and diurnally-averaged soil moisture in DA_100_100 is due to the fact that the soil cannot stay saturated and thus the soil model will produce an instantaneous runoff to bring back soil moisture to the field capacity. However this has no effect on the evaporation as it does not change for soil moisture values larger than the field capacity (see Fig.7 of the manuscript). We will clarify this aspect in the manuscript.
When deciding the domain size we wanted to reduce as much as possible the dynamical contribution of advection on precipitation, that is the spurious effect of the front collision on precipitation. As showed in Fig. 6 the collision of the two fronts in the middle of the dry patch,

because of the periodic boundary condition, enhances convergence, uplift and thus precipitation. Since such effect can alter the interpretation of the results we wanted to delay the diurnal front collision as much as possible, while keeping the computation costs affordable for running several sensitivity experiments. That's why we settled on a domain which is 400x100 km² big.

**2. The simplified model**

1) **Assumptions:** According to Section 4, the authors assumed that "Ewet does not depend on Ï¸Tdry" (Pg. 13, line 1) and "evaporation over the dry patch does not depend on the soil moisture of the wet patch" (Pg. 14, line 1). These two assumptions are needed to get the key results (Eqs. 10, and 15), but not clearly justified. When either Ï¸Tdry or Ï¸Twet varies, should not precipitation over the wet or dry patches change, which in turn impact Ewet or Edry through the impact on soil moisture therein? On Pg. 11 (lines 2-7), it is assumed that the advection of water vapor and hydrometeors is mainly constrained in the boundary layer. As shown in Fig. 2, however, the return flow at ~1-3 km is not negligible. Could you please justify this assumption further?

**Response**: There is no dependency of $E_{wet}$ on $\phi_{dry}$ because in Eq. 4, and more generally in the Budyko formulation, evaporation over a surface depends on the local soil moisture, in this case $\phi_{wet}$. Furthermore, our formulation of the evaporation considers a soil moisture constant in time, as explained in the answer to the previous comments. This is well justified as we only consider one diurnal cycle: over this period precipitation is expected to change soil moisture only marginally and thus not to change evaporation appreciably. Regarding the return flow it should be noted that in our simulation this branch of the circulation has a much weaker intensity (in terms of zonal velocities at least 50% less) and lasts just for a few hours. For these reasons we consider it as negligible when developing the idealized model.

2) **Derivations:** Please provide more details on how to approximate Eq. 5 to get Eq. 6, and how Eq. 7 is obtained. To get Eq. 8, it seems that one has to assume the vertical extent of moistening process due to latent heat flux, Hmoist, is the same over the dry and wet patches. Is Hmoist related to turbulent eddies? If so, this assumption can be problematic because low level temperature differences are up to ~4 K between the dry and wet patches (Fig. 2), where sensible heat flux differences can reach 280 W/m2 (Pg. 5, lines 5-6).

**Response**: We decided to revise section 4.1 and 4.2 and specifically to remove Eq. 6 since it contained an ambiguous notation. Instead we decided to include the approximation of specific humidity (eq. 7) and then proceed to explain how to obtain Eq. 8. The latter equation assumes that the vertical extent of the moistening process $H_{moist}$ is the same as the vertical extent of the breeze circulation $H_{front}$. We agree with the reviewer that the two are not exactly the same. To check the validity of this assumption we computed these two heights from the simulations. $H_{moist}$ was computed as the height of the PBL over the wet patch, diagnosed with the bulk Richardson number method (see Seibert et al., 2000). $H_{front}$ instead was computed as the height at which the zonal pressure anomaly ahead of the front reaches 0 (see Rochetin et al., 2017). Although slight differences up to 300-500 m were present at some time instants, the two variables showed similar values. As the goal is to develop a simplified model that only retains the main drivers of precipitation variability, we think that the assumption is well justified. Furthermore, it should be noted that, since $H_{moist}$ does not depend on $\phi_{dry}$, including its effect won't change the results, i.e. the fact that the derivative of precipitation does not depend on $\phi_{dry}$. We included this information in section 4 of the manuscript.

3) **Comparison to Lintner et al. (2013)** As noted in the article (Pg. 10, lines 26-30), feedbacks between the land-surface and atmosphere are neglected in the simplified model after taking evaporation as Eq. 4. Consequently, it is not unexpected that soil moisture can be irrelevant to

precipitation change in the simplified model. If the formulation of evaporation in Lintner et al. (2013), where land-atmosphere interactions are considered, is used in the derivation, will the theoretical model proposed here still be valid?

**Response**: We have to disagree with the reviewer. Within the framework of our simplified model precipitation changes are independent of soil moisture as the derivative of precipitation with respect to soil moisture does not depend on the latter. As explained in the manuscript this is an effect not only of the linear dependency of evaporation on soil moisture but also of the constant front velocity. Note that the idealized model does not consider explicit land-surface interactions but is in good agreement with the results of the simulations which are coming from a coupled land-atmosphere model. Thus, we don't think that the results would differ much in case the land-atmosphere interactions would be considered as long as one diurnal cycle is considered (see also our response to comment 1 above).

4) **Precipitation efficiency associated with evaporation and advection** Could you please elaborate further on why precipitation efficiency is independent of soil moisture? Although the authors showed that precipitation efficiency associated with advection is independent of evaporation using the extreme case DA_20_100, where Edry is negligible, it is not clear on why precipitation efficiency associated with evaporation is independent of soil moisture. Overall, it is hard to evaluate the simplified model according to how it is presented. The conclusion that precipitation change over a heterogenous surface is independent of soil moisture can be an artifact that land-atmosphere interactions are eliminated in the theoretical model.

**Response**: From the physical point of view, soil moisture controls directly evaporation but not precipitation. The control of soil moisture on evaporation is already included in Eq. 4. The efficiencies then describe the processes that take place in the atmosphere which convert moisture sources (advection and evaporation) into precipitation. They are used to represent the fact that two different atmospheric states will produce different precipitation amount, even though the soil moisture and evaporation can be identical. For this reason, the efficiencies should not be defined as functions of soil moisture.

The conclusion that precipitation changes over a heterogeneous surface are independent of soil moisture is related to the assumptions made in the idealized model: as long as the front propagation velocity does not depend on soil moisture and the evaporation is a linear function in soil moisture our results won't be affected. Moreover, as shown in Figs. 8 and 9 of the manuscript, our assumptions are justified as our model, despite its simplicity, is able to reproduce the simulation results fairly well. We clarified these aspects in the conclusions and in section 4.

**3. Writing**
The manuscript requires an editorial revision to correct wording issues. Some sentences are either awkward or redundant. For example, ". . ., in a nutshell, . . ." (Pg. 6, line 4), ".. thanks to the previous section . . ." (Pg. 12, line 12) and etc. can be removed.

**Response**: we removed these ambiguous sentences.

**Minor Comments**
1. Are equations 5 and A2 written correctly as advection? It is also unusual to have dot product between a scalar (qtot) and a vector (ufront or v).

**Response**: We modified the equations by removing the dot and using vector notation. Otherwise the equations are correctly written.

2. Pg. 2, lines 16-17: Please clarify further on Guillod et al. (2015), what does ". . . a negative spatial coupling coexists together with a positive temporal coupling" mean and indicate?

**Response**: We added a clarification sentence: " That is, areas drier than their surrounding (spatial component) but wetter than the climatological value (temporal component) may receive more precipitation than other ones"

3.Pg. 3, line 25: Why a rectangular domain can limit computational cost?
**Response**: This sentence was meant to represent a comparison between a squared domain (which in our case would be 400x400 km$^2$ big) and our chosen rectangular domain (400x100 km$^2$) which contains less grid points and is thus less expensive to run simulations on.

4.Pg. 9, lines 6-7: Please provide relevant evidences on ". . . several secondary events develop due to the waves propagating away from the collision".
**Response**: We agree that using the term"wave" was not appropriate so we rephrased this part to read "In the ID_20_100 case strong precipitation events with local maxima of 10 mm h$^{-1}$ are produced in the center of the patch after the fronts' collision and several secondary events develop due to the fronts propagating again away from the collision. "

5.Fig. 2: It can be better to show wind as vectors, rather than contours.
**Response**: We revised Fig. 2 to ease its interpretation. Now winds are displayed through vectors and not contours.

6.Fig. 9: Change "$\partial$Pdry/Ï¸Tdry" as "$\partial$Pdry/$\partial$Ï¸Tdry".
**Response**: We thank the reviewer for spotting this typo, which we promptly corrected.

7.Table 1: Change "Name" as "Experiment".
**Response**: Corrected

---

## Author Response (AR2)

**General Comments**

I would like to thank the authors for the replies. As described in the following, I am asking for several minor revisions easy to implement and do not need to review this manuscript again for final acceptance.

> We thank again the reviewer for the comments. In the following we are attaching our responses to these final minor comments.

**Major Comments**

1. The manuscript is improved after the clarification that the proposed simple model only works if evaporation depends linearly on soil moisture and the propagation speed of the front associated with heterogeneity induced mesoscale circulation does not vary much with surface heterogeneity gradient. Besides in the conclusions, these limitations should be emphasized in the abstract as well, as it is not clear on how much and how often these two conditions are held in nature.

> We have to partially disagree with the reviewer. First of all, the linear dependence of surface evaporation on soil moisture has been shown to be a fairly good approximation for intermediate values of soil moisture (see e.g. Schwingshackl, C., M. Hirschi, and S.I. Seneviratne, 2017, https://doi.org/10.1175/JCLI-D-16-0727.1). Second, the acceleration of the front due to presence cold pools will be observed as long as precipitation occurs (Rieck, M. , Hohenegger, C. and Gentine, P. ,2015, https://doi.org/10.1002/qj.2532). Many studies that attempted to find a scaling law for the propagation speed of thermally-driven mesoscale circulations often did not consider the production of precipitation. For this reason, they found a strong dependence of the propagation speed to the surface state (in particular sensible heat fluxes). We already discussed these two "limitations" of the conceptual models in section 4 and in the conclusions, thus we think that adding another line in the Abstract would be redundant as the abstract only needs to contain the main features of the manuscript.

2. As indicated by Fig. 5, soil moisture in the numerical simulations does change strongly with time over the dry patch if not averaged over time. It will be better to explicitly claim the rainfall here refers to accumulation over a diurnal cycle. Otherwise, a basic assumption in the simple model, i.e., soil moisture does not change with time, is invalid. This clear statement about the diurnal time scale will also further clarify the assumption that water vapor from evaporation and advection may not be well-mixed.

> Figure 5 does not show soil moisture as function of time and, as showed by the Table attached to the previous response to the Reviewer, the value of soil moisture averaged over the dry patch actually does not change appreciably over the diurnal cycle, as the initial and final values are similar.

3. In the first submission, the author noted that "Our approach in deriving the theoretical model directly neglects the feedback between the land-surface and the atmosphere, as evaporation does not implicitly depend on the near-surface atmospheric specific humidity but only on soil moisture" (Pg. 10, lines 27-29). Although the simple model agrees with the numerical simulations well, probably because evaporation is occasionally linearly dependent on soil moisture under the idealized configuration, it will be informative to have some discussions on the caveats related to this simplification. This can be added in the last section of the manuscript.

> The conditions under which our simple conceptual model is valid are already mentioned multiple times in the manuscript so we decided to not add a repetitive discussion at the end of the last section.

**Minor Comments**

1. Pg. 11, lines 8 - 11: The interpretation on how convective updrafts are less affected by dry air entrainment in the ID cases than in the DA experiments is not sufficient. Besides the impact from moisture difference between the updrafts and the environment, their MSE differences also depend on the vertical profiles of convective mass flux which is associated with entrainment rate.

> We computed the entrainment rate by considering the ratio between the vertical derivative of updraft MSE and the difference in MSE between updraft and the environment. When integrating the vertical profile of entrainment, larger values are obtained in the DA case with respect to the ID case, given the wider extension of the vertical column of the average updraft. Thus, this confirms our hypothesis that, in the DA case, the updraft are more influenced by the entrainment of dry air. This eventually suppress convection and the production of precipitation.

2. It may be better to switch the two panels in Fig. 7, as the DA cases are discussed before the ID cases in most part of the manuscript.

> Following the suggestion of the reviewer, we switched the order of the two panels in Fig. 7.

3. Pg. 14, line 14: Please briefly explain why it is an "upper bound".

[revised manuscript text omitted]